# OPIC: Enhancing Language Model Merging via Optimizing In-Context Capability

**Jie He** [1]  **Weidong Bao** [1]  **Chao Chen** [1]  **Zhengyi Zhong** [1]  **Shuai Zhang** [1]  **Ji Wang** [1]

## Abstract

Task-vector–based model merging enables low-cost, training-free multi-task learning for large language models, but suffers from performance degradation compared to individually fine-tuned models. Prior mitigation strategies largely rely on validation data for costly hyperparameter tuning, limiting both interpretability and practicality. We therefore propose OPIC, an evolutionary optimization–based model merging framework. Our preliminary experiments reveal that the degradation of in-context learning (ICL) capabilities exhibits a strong correlation with performance deterioration. Motivated by this insight, we formulate model merging as an optimization problem with ICL preservation as the objective. OPIC introduces a hierarchical refinement operators and optimizes it using self-generated data, effectively eliminating the reliance on external validation sets. Experimental results demonstrate that OPIC achieves an average performance retention of 80.73%, outperforming SOTA methods and improving by up to 11.1% over recent validation-free approaches. In addition, OPIC is compatible with existing merging pipelines, offering a new alternative solution for deploying without validation dependencies. Code is available at: https://github.com/illusion-hj/OPIC

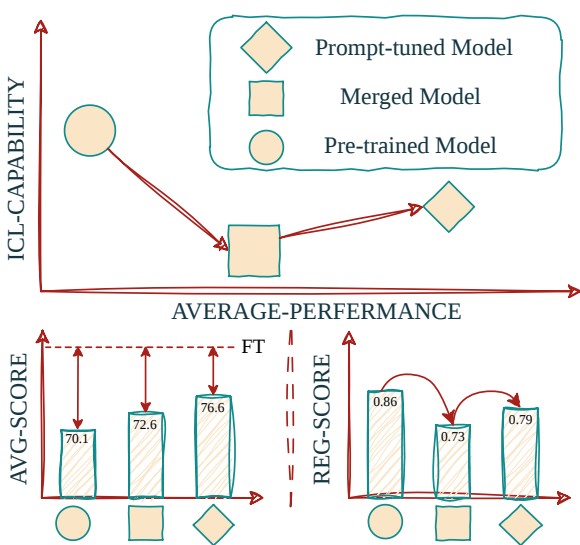

*Figure 1.* Evaluating the correlation between in-context learning (ICL) capability and merging performance. Four tasks are merged into Qwen1.5-7B via Task Arithmetic, followed by Soft Prompt Tuning with DROID to enhance ICL without modifying parameters or task knowledge. Performance is assessed on REGBench and downstream tasks.

## 1. Introduction

**Background.** Although pre-trained models and fine-tuning techniques (Kenton et al., 2019; Shnarch et al., 2022; Radford et al., 2021; Zhong et al., 2025) has achieved remarkable performance across various domains(Zhuang et al., 2020; Bommasani, 2021), their broader application is lim-

[1]National Key Laboratory of Big Data and Decision, National University of Defense Technology, Changsha, Hunan, China. Correspondence to: Ji Wang <wangji@nudt.edu.cn>.

*Proceedings of the 43rd International Conference on Machine Learning*, Seoul, South Korea. PMLR 306, 2026. Copyright 2026 by the author(s).

ited by data privacy constraints and the growing demand for better generalization in real-world scenarios(Wolf et al., 2019). Meanwhile, traditional multi-task learning methods (Raffel et al., 2020; Sanh et al., 2021) suffer from high training costs and data reliance from multiple tasks(Fifty et al., 2021; Dong et al., 2024; Lu et al., 2023; Qiu & Yang, 2024). Against this background, model merging methods based on *task vectors*, which represent the difference in parameter space derived from the fine-tuned model, have emerged as a new promising solution(Ilharco et al., 2022). The core rationale behind this approach is to use task vectors to capture the knowledge embedded in fine-tuned models and integrate multiple such vectors into the pre-trained model, thereby enabling efficient multi-task adaptation without additional training(Jin et al., 2022; Matena & Raffel, 2022).

**Shortcomings and Challenges.** However, a key drawback of such methods is inter-task interference during merging, which degrades the merged model's performance on indi-

vidual tasks compared to the original fine-tuned models. Recent research focuses on overcoming this difficulty in the *data-free* fine-tuning manner(Sun et al., 2025a;b). The typical method is to achieve alignment by eliminating parameter redundancy and direction conflicts(See Appendix A for more details). Despite recent progress, existing research still has two major limitations, *explanation-limited* and *validation-dependent*.

- At the theoretical level, prior works predominantly attribute performance degradation to inter-task conflicts, focusing on resolving this through parameter-level alignment. However, these considerations overlook the potential impact of merging on the underlying pre-trained model, which motivates our first challenge: *Can we identify the critical factors behind performance degradation from a novel perspective?*
- At the methodological level, most methods need to rely on the validation set to determine hyperparameters, which has a great impact on performance. This dependence severely limits their usefulness in real-world scenarios and goes against the original intention of model fusion. This leads to the second challenge, *how to design an validation-free algorithm to achieve efficient model merging?*

**Proposed solution.** We shift the perspective from task space to the property space of large language models to investigate the key factors underlying performance degradation. As illustrated in Figure 1,We observe that model merging degrades in-context learning (ICL) capabilities. Notably, restoring this capability without altering the merged weights consistently leads to improved downstream performance. This phenomenon inspires us that ICL capabilities may be a key indicator for maintaining performance in the merger process. A comprehensive analysis of this pattern is provided in the Section 3. Leveraging this insight, we introduce OPIC, an merging framework based on **op**timization algorithms and **i**n-**c**ontext learning. Instead of relying on real validation data, OPIC utilizes synthetic data (REGBench) to measure ICL retention. With the target of ICL, we model the merging processes as an optimization problem, introduce the hierarchical merging operator as a variable, and solve it using optimization algorithm.

**Contributions.** In summary, our main contributions are summarized as follows:

- **A Novel Perspective on Understanding Performance Degradation of Model Merging.** Moving beyond prior parameter-level analyses, this work investigates performance degradation in model merging from the perspective of model capabilities. We identify the maintenance of contextual capability is a key indicator to ensure the performance of model merging, offering a novel and more explanatory framework for understanding performance degradation on merging process.

- **A Validation-Free model merging Approach.** Unlike existing approaches that rely on validation datasets for hyperparameter tuning, our method leverages artificially generated data as a proxy evaluation set, completely removing the dependence on real data.

- **A Practical Integration Mechanism.** Our OPIC method can be seamlessly integrated into existing model merging pipelines. Empirical results demonstrate that it preserves their performance while eliminating reliance on validation datasets, significantly enhancing practicality.

## 2. Related Work

To address the performance degradation caused by task conflicts, we categorize existing model merging methods according to how the merged knowledge is stored and used during inference. Test-time adaptation and data-free methods ultimately produce a single merged model, whereas MoE-like methods retain multiple models or task-specific components and rely on a router to perform model inference. Based on this view, three mainstream approaches have been proposed: Test-time adaptation, MoE-like methods and Data-free. Although Test-time adaptation and MoE-like methods have achieved certain success, their practical applicability is limited due to challenges such as data privacy concerns, additional storage requirements, and the lack of parallelism in the MoE model(Cheng et al., 2025). Therefore, this paper primarily focuses on data-free model merging.

**Test-time adaptation Model Merging.** Test-time adaptation model merging utilizes a portion of unlabeled test data to resolve task conflicts. For example: Yang et al. proposed AdaMerging(Yang et al., 2023), which utilized entropy minimization on unlabeled test samples as a heuristic objective function to learn the merging coefficients. Despite the promise of these methods, a critical limitation lies in their reliance on access to test data, which may not be feasible in practice.

**MoE-like Model Merging.** MoE-like model merging is achieved by storing specific knowledge for each task and performing hybrid expert routing. Twin-Merging shows that task-specific models often contain hybrid knowledge, where expertise in one model may be exclusive or even detrimental to other models. It argues that the fusion and organization patterns of shared and private task-specific knowledge are beneficial for ensuring merged model performance(Lu et al., 2024). Huang et al. (2024) proposed EMR-Merging, which achieves model merging by designing a unified public task vector and a lightweight task-specific modulator(Huang et al., 2024). Although these methods effectively mitigate interference in model merging, they require additional memory overhead to store multiple task-specific components, as well as additional test data to train the router.

**Data-Free Model Merging.** Data-free model merging aims to combine different expert models without the need for raw training data. Task Arithmetic(Ilharco et al., 2022) represents the most classic method that edits models by simply applying arithmetic operations to task vectors. Based on this method, some representative works are as follows: Yadav et al. (2023) proposed Ties-Merging to resolve this challenge by three novel steps(Yadav et al., 2023). DARE(Yu et al., 2024) and PCBMerging(Du et al., 2024) resolve the merging interference by dropping parameters and unscaling operations. Wang et al. (2024) introduced Consensus Merging, which enhanced the overall performance of existing model merging methods by removing the selfish and catastrophic weights(Wang et al., 2024).

While the data-free methods are able to run without raw training data, they still rely on a set of validation data to evaluate performance and use it as an optimization goal to determine the optimal hyperparameters. Recent studies have attempted to cast the efficient alignment between the merged model and the expert models as the optimization objective of task vectors, thereby eliminating the need for test data.(Wei et al., 2025). In this paper, we introduce a new validation-free approach, inspired by the emergent capabilities of large language models.

## 3. Preliminary and Motivation

This section first gives the basic paradigm of model merging, and then uses pre-experiments to further explore the correlation between in-context learning (ICL) capability and merging performance, laying the foundation for our subsequent method design.

### 3.1. Preliminary

A simple yet effective merging strategy is weight averaging, or model soups. For large language models, this idea is exemplified by Task Arithmetic(Ilharco et al., 2022), where task-specific knowledge is represented as task vectors. Given a pretrained model with parameters $\theta_0$ and task-specific models $\theta_k$, task vectors are defined as $\tau_k = \theta_k - \theta_0$, and the merged model is obtained by:

$$\theta_{\text{merged}} = \theta_0 + \lambda \sum_k \tau_k, \tag{1}$$

where $\lambda$ controls the contribution of task-specific updates.

### 3.2. Motivation: The Correlation Between ICL and Performance

In this work, we propose shifting the perspective from parameter statistics to behavioral capabilities to investigate the key factors of performance degradation. Studies have shown that the performance of large language models on down-stream tasks is closely related to their emergent abilities, such as complex logical reasoning, code generation, multilingual translation, and even a degree of creativity(Berti et al., 2025; Krakauer et al., 2025). Among these emergent abilities, ICL is considered a foundational mechanism for enabling building complex reasoning chains(Baldauf et al., 2007; Perera et al., 2013; Herrmann et al., 2025). In light of this, we hypothesize that *the loss of in-context learning capability might be the key factor underlying performance degradation during model merging*. To validate this hypothesis, we conduct a preliminary study analyzing the relationship between ICL performance and downstream task performance during the merging process.

**Setup.** We apply Qwen1.5B as the backbone network, fine-tune and evaluate it on 4 tasks(ANLI, MC_TACO, COQA and MATHQA datasets), using Task-Arithmetic method for fusion, and search for $\lambda$ with a step size of 0.1 from 0 to 1.

Measuring ICL Capability: To accurately quantify the model's ICL capability while avoiding reliance on real-world datasets, we adopt REGBench(Akyürek et al., 2024) as the evaluation dataset, following the work of Ekin Akyürek et.al. We use **greedy decoding accuracy** to quantify the model's ICL ability. This metric measures whether the model's predicted next token is valid under the given language. Specifically, for all prediction positions in the test set, the model's ICL ability can be formulated as:

$$
\begin{aligned}
\text{Accuracy} &= \frac{1}{N_T} \sum_{i=1}^{n} \sum_{j=1}^{|d^{(i)}|} \mathbb{I}\left[ \hat{x}_j^{(i)} \in V\left( d_{<j}^{(i)}, L^{(i)} \right) \right] \\
V(c, L) &= \{x' : L(x' \mid c) > 0\} \\
\hat{x}_j^{(i)} &= \arg\max_x p_\theta(x \mid d_{<j}^{(i)}),
\end{aligned}
\tag{2}
$$

where $V(c, L) = \{x' : L(x' \mid c) > 0\}$ is a validation function that checks if a predicted token is valid under a given language and context. $p_\theta(\cdot \mid \cdot)$ is the probability distribution of the next token predicted by the neural sequence model with parameters $\theta$.

Measuring Task Performance: To eliminate the baseline shifts and focus on relative differences, we subtract the minimum value in performance and normalize using a fine-tuned model as the denominator (in some test sets, the worst score of performance is not 0, e.g., four-choice questions, random answers still have an expected accuracy of 25%). The overall performance of the fusion model is calculated by the equation (3).

$$\text{AVG Score} = \frac{1}{T} \sum_{t=1}^{T} \frac{\text{Score}[f(\theta^*)] - \min_{k \in 1,\dots,T} \text{Score}[f(\theta_k)]}{\text{Score}[f(\theta_t)]}, \tag{3}$$

where $\theta^*$ represents the fusion model, and $\theta_t$ represents the fine-tuned model.

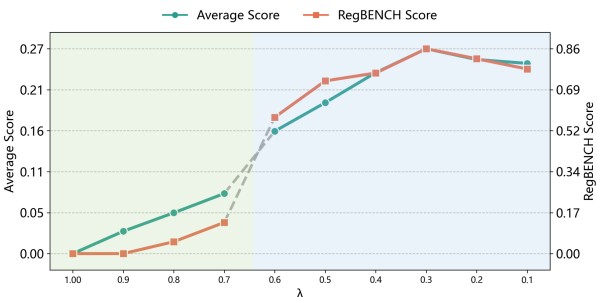

*Figure 2.* The performance scores on both the REGBench and MathQA test sets vary with the choice of parameters $\lambda$.

**Observations:** The results of the pre-training experiment are shown in the Figure 2. From the figure, we can observe two phenomena:

- **The Importance of Hyperparameters:** Across the continuous parameter variations, a distinct performance gap is observed between the performance curve and the contextual capability curve. This phenomena indicates the critical impact of hyperparameter settings on the performance of model merging. Precisely because hyperparameters need to be determined through verification data and highly correlated with performance, they significantly limit the effectiveness and applicability of mainstream methods such as TIES and DARE.
- **Strong Positive Correlation:** To the left of the gap, the model's contextual capability is almost entirely lost (approaching 0), with its relative performance also near zero. The restoration of contextual capability is accompanied by an improvement in performance. To the right of the gap, a significant positive correlation exists between the model's ICL ability and the preservation of its specialized capabilities. This demonstrates that maintaining contextual capability is crucial during the model fusion process, and that stronger contextual ability often correlates with superior performance of the merged model.

The above experimental phenomena effectively support our hypothesis. *In addition, it inspired us that leveraging self-generated data to monitor the performance of the merging process is a viable path and cleverly bypasses the dependency on test datasets.*

## 4. Methodology

Based on the motivation, we propose OPIC. The core idea of our method is to leverage the positive correlation between contextual capability and the performance of the merged model, using contextual capability as the optimization objective and applying an optimization algorithm to iteratively refine the model merging process. Specifically, we use self-generated data (REGBench) to evaluate the preservation of

contextual capability during model merging and adopt it as the optimization target. We also design a hierarchical refinement operator to modify task vectors and employ a genetic algorithm to achieve adaptive parameter optimization and model merging. The overall framework of our algorithm is illustrated in Figure 3.

### 4.1. Refinement Operator Design

Research by Enneng Yang et al.(Yang et al., 2023) demonstrates that hierarchical fine-grained weighting can integrate knowledge from multiple tasks more efficiently and effectively. In light of this, we propose a hierarchical framework for task vector fusion. The core innovation lies in introducing three learnable parameters to precisely control the fusion process:

- Inter-layer Weight Matrix ($A \in \mathbb{R}^{M \times N}$): A matrix designed to weight the $N$ task vectors differently across layer depths. Where $M$ is the number of model layers, each column of $A$ corresponds to the cross-layer weighting scheme for a specific task.
- Task Scaling Vector ($B \in \mathbb{R}^{1 \times N}$): This vector applies an independent scaling factor to each task, globally adjusting the contribution of the entire corresponding task vector after inter-layer weighting.
- Global Scaling Scalar ($C \in \mathbb{R}$): A scalar constant used to globally scale the fused task vector.

### 4.2. Optimization Objective

Our goal is to identify an optimal set of parameters $(A^*, B^*, C^*)$ that maximizes the accuracy of the fused model on the REGBench test dataset. Let $\mathrm{Acc}(\theta_{\mathrm{fused}}, \mathcal{D}_{\mathrm{REGBench}})$ denote the accuracy of a model with weights $\theta_{\mathrm{fused}}$ evaluated on the REGBench dataset $\mathcal{D}_{\mathrm{REGBench}}$. This objective can be formally expressed as the following constrained optimization problem:

$$(A^*, B^*, C^*) = \underset{A,B,C}{\arg\max} \mathrm{Acc}\left(\theta_{\mathrm{fused}}, \mathcal{D}_{\mathrm{REGBench}}\right), \quad (4)$$

where the calculation of $\mathrm{Acc}\left(\theta_{\mathrm{fused}}, \mathcal{D}_{\mathrm{REGBench}}\right)$ is the same as the Equation (2).

### 4.3. Overall Workflow of OPIC

Stage 1: Get the Task Vectors. Let $\theta \in \mathbb{R}^d$ denote the weight vector of the pre-trained base model. For each task $k \in 1, 2, \ldots, N$, let $\theta_k \in \mathbb{R}^d$ represent the weight vector of the model fine-tuned on that task. The task vector is obtained by the following equation:

$$\tau_k = \theta_k - \theta. \quad (5)$$

Stage 2: Obtain Refinement Operators through Optimization. As shown in Algorithm 1, we target the contextual

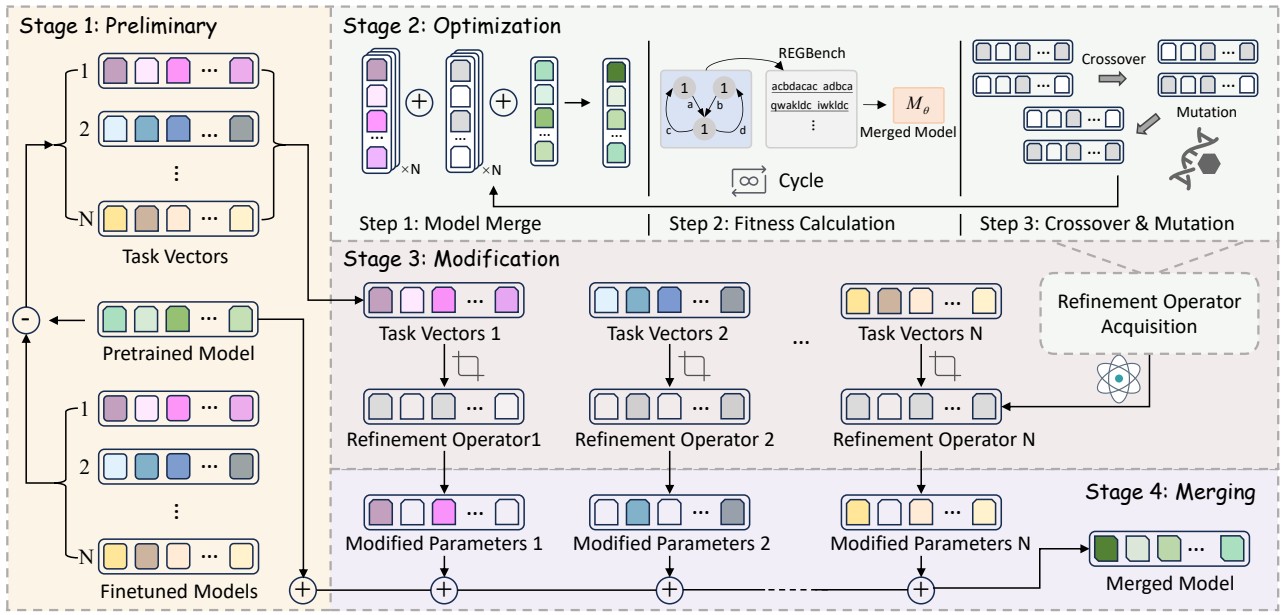

*Figure 3.* Framework of the OPIC Algorithm: The proposed framework consists of four stages: Preliminary, Optimization, Modifacation, and Merging. Task vectors are first extracted from fine-tuned and pre-trained models, then refined by a refinement operator optimized via an evolutionary algorithm on the self-generated REGBench dataset to preserve ICL, and finally merged into the pre-trained model through an element-wise product to obtain the final model.

---

**Algorithm 1** Refinement Operator Acquisition

---

**Input**: Regbench test data, Pre-train and Fine-tuning models
**Parameter**: $A, B, C$
**Output**: Refinement Operator

1: Initialize evolutionary algorithm population.
2: **while** not termination **do**
3:     **for** each individual $i$ in population **do**
4:         Modify the task vector based on $i$
5:         Run model merging based on Equation (8);
6:         Evaluate fitness based on Equation (2);
7:     **end for**
8:     Evolve population by crossover and mutation
9: **end while**
10: Save the best solution.

---

capability and adopt optimization algorithms to refine the corresponding refinement operator. To achieve an adaptive search for the algorithm's parameters, we employ the Strengthen Elitist GA Algorithm (SEGA)(Liang & Leung, 2011) to solve the aforementioned optimization objective, which is a meta-heuristic global search method that simulates the process of natural selection.

Stage 3: Modified with the Refinement Operator. Firstly, get the final layer-task weight matrix $W$:

$$W = B \odot A, \qquad (6)$$

where $B \odot A$ denotes the broadcasted Hadamard product,

such that $W_{i,k} = B_k \cdot A_{i,k}$, where $i \in \{1, \dots, M\}$ is the layer index and $k \in \{1, \dots, N\}$ is the task index.

Next, the set of task vectors $\{\tau_k\}_{k=1}^N$ is aggregated using a weighted sum with the matrix $W$, producing an intermediate fused vector. Finally, this result is then scaled globally by the scalar $C$:

$$\tau_{\text{fused}} = C \cdot \left( \sum_{k=1}^{N} \sum_{i=1}^{M} W_{i,k} \cdot \tau_k^{(i)} \right), \qquad (7)$$

where $\tau_k^{(i)}$ denotes the sub-vector of task vector $\tau_k$ corresponding to the weights of the $i$-th model layer.

Stage 4: Merging Base and Fused Components. Integrating the expressions above, the final weights of the fused model are given by:

$$\theta_{\text{fused}} = \theta + \tau_{\text{fused}}. \qquad (8)$$

### 4.4. Integration with Existing Merging Algorithms

Our method follows the same fundamental paradigm as existing merging algorithms—namely, the refinement of task vectors and the optimization of fusion parameters. Therefore, our algorithm can be seamlessly incorporated into various existing model merging pipelines, according to the following basic procedure:

1. Composition of Modification Operators: First, use the existing algorithm to modify the task vectors. Then, apply our proposed operator to scale the results from a model-

layer perspective. Taking DARE as an example, the fused task vector can be expressed as:

$$\tau_{\text{fused}} = C \cdot \left( \sum_{k=1}^{N} \sum_{i=1}^{M} W_{i,k} \cdot \text{DARE}(\tau_k^{(i)}) \right). \quad (9)$$

2. Replacement of Parameter Search Method: Replace the conventional approach of using a test dataset for evaluation and grid search, by using REGBench as the test set and a genetic algorithm employed to iteratively optimize the hyperparameters.

Integrating our method with other methods has two advantages: 1. Overcoming the dependency on the test dataset by using REGBench and using the following capabilities as evaluation indicators; 2. Adaptive optimization of hyperparameters is implemented through genetic algorithms.

# 5. Experiments

In this section, we first describe our experimental setup. Then, we present our main results. We also provide ablation studies and discussions for a thorough analysis.

## 5.1. Experimental Setup

**Datasets and Models.** We utilize the ANLI, COQA, MC_TACO and MATHQA dataset to conduct experiments, which involves mathematical operations, code compilation, and basic abilities. In addition, the dataset ARC, CROWS_PAIRS and MMLU are used for generalization evaluation. We validate our method across mainstream LLMs with distinct architectures and parameter scales, specifically Qwen2.5-1.5B, Qwen2.5-3B, and Llama3.2-3B. A more detailed description and sources of citations for the datasets and models are provided in Appendix C.3.

**Baselines.** We compare our method against several representative and state-of-the-art approaches in the field of the data-free model merging, including Task Arithmetic*[ICLR 2023]*(Ilharco et al., 2022), TIES*[NeurIPS 2023]*(Yadav et al., 2023), DARE*[ICML 2024]*(Yu et al., 2024), CABS*[ICML 2025]*(Yang et al., 2025) and DOGE*[ICLM 2025]*(Wei et al., 2025). For details on the baselines, please refer to Appendix C.2.

**Other Implementation Details.** Details on the grid search strategy are provided in Appendix C.4. Training details, hardware configuration and evaluation platform are described in Appendix C.5 ,C.6 , respectively.

**Metrics.** Given the diversity of evaluation metrics across datasets, we employ a normalization strategy to unify model score through dividing the model's performance by that of

the fine-tuned baseline, as defined below:

$$\text{Normalized Score} = \frac{\text{Score}[f(\theta^*)]}{\text{Score}[f(\theta_t)]}, \quad (10)$$

where $\theta^*$ represents the fusion model, and $\theta_t$ represents the fine-tuned model.

## 5.2. Main Results

**Performance comparison.** Table 1 presents the overall performance of each method on the test set. We calculate the score using Equation (10), with the score before normalization in parentheses, and the detailed calculation metrics for each dataset are shown in the Appendix C.1. The best performance is shown in bold, and the second-best is underlined. From the table, we can observe the following:

1. Our method achieves the best average performance, which validates the effectiveness of our theory and approach. Moreover, when examining the performance across individual datasets, the gap between our method's best and worst scores is smaller, at 32.5%, compared to 34.0% for Task Arithmetic, 41.2% for TIES, 33.3% for DARE, 47.2% for DOGE, demonstrating better robustness.

2. Under the validation-free setting, our method achieves substantial performance improvements, outperforming the state-of-the-art approach (DOGE) across all test sets. The largest gain reaches 30.1% (on the ANLI dataset), with an average improvement of 11.11%.

3. The average performance of TIES and DARE is inferior to that of simple Task Arithmetic. This finding corroborates the theory in (Wang et al., 2024) and also echoes the shortcomings of existing research that we pointed out in the Introduction: namely, theories based on parameter redundancy and direction conflicts cannot adequately explain the performance degradation in model merging, and the effectiveness of such methods has significant limitations.

**Extension to Other Models.** Furthermore, we conducted more comprehensive supplementary experiments on models of different scales (Qwen2.5-3B) and different architectures (Llama3.2-3B), the detailed results are presented in Table 2. in experiments using Qwen2.5-3B, compared with other state-of-the-art methods, our approach consistently maintains a leading performanceas, achieving the highest average performance among all methods. Notably, on the MathQA dataset, our method even surpasses the corresponding fine-tuned model, demonstrating both its effectiveness and robustness.

In experiments based on LLaMA-3B, our method also achieves the best overall performance, improving by 11.98% over the weakest baseline. Although DOGE exhibits unexpectedly strong performance on the COQA and MC_TACO datasets, its inferior and unstable results on other datasets

*Table 1.* Performance Comparison of the Fusion Models Base on Qwen2.5-1.5B.

| Method | Validation Req. | ANLI(↑) | COQA(↑) | MC_TACO(↑) | MATHQA(↑) | Mean(↑) |
|---|---|---|---|---|---|---|
| Pre-trained | – | 0.414 | 0.420 | 0.627 | 0.345 | 78.02% |
| Individual | – | 0.634 | 0.630 | 0.953 | 0.417 | 100% |
| Task Arithmetic | True | **97.9%** (0.621) | 63.9% (0.403) | 74.9% (0.714) | 81.5% (0.340) | 79.56% |
| TIES | True | 80.0% (0.507) | 58.9% (0.371) | 70.8% (0.675) | **100.1%** (0.421) | 77.63% |
| DARE | True | 71.8% (0.455) | **66.2%** (0.417) | 75.8% (0.723) | 99.5% (0.415) | 78.32% |
| CABS | True | 68.1% (0.432) | 58.5% (0.369) | 73.2% (0.698) | 94.5% (0.698) | 73.59% |
| DOGE | False | 67.4% (0.427) | 63.9% (0.403) | 50.0% (0.477) | 97.2% (0.406) | 69.62% |
| OPIC(Ours) | False | 97.5% (0.618) | 65.0% (0.410) | **77.4%** (0.738) | 83.0% (0.346) | **80.73%** |

*Table 2.* Performance of the Fusion Models Base on Qwen2.5-3B and Llama3.2-3B.

| Method | Validation Req. | ANLI(↑) | COQA(↑) | MC_TACO(↑) | MATHQA(↑) | Mean(↑) |
|---|---|---|---|---|---|---|
| Qwen2.5-3B | | | | | | |
| Pre-trained | – | 0.465 | 0.655 | 0.711 | 0.374 | 77.87% |
| Individual | – | 0.731 | 0.678 | 0.983 | 0.474 | 100% |
| Task Arithmetic | True | 69.6% (0.509) | 94.9% (0.644) | 79.8% (0.784) | 99.8% (0.473) | 86.02% |
| TIES | True | 68.0% (0.497) | 96.0% (0.651) | 74.8% (0.735) | 98.4% (0.467) | 84.31% |
| DARE | True | 67.9% (0.496) | 95.5% (0.648) | **79.9%** (0.785) | **101.2%** (0.480) | 86.12% |
| CABS | True | **71.7%** (0.524) | 89.9% (0.609) | 78.3% (0.769) | 97.8% (0.464) | 84.42% |
| DOGE | False | 58.7% (0.429) | 95.1% (0.645) | 77.7% (0.763) | 94.5% (0.448) | 81.50% |
| OPIC(Ours) | False | 69.1% (0.505) | **96.6%** (0.655) | 79.3% (0.779) | 100.2% (0.475) | **86.29%** |
| Llama3.2-3B | | | | | | |
| Pre-trained | – | 0.335 | 0.649 | 0.415 | 0.347 | 62.31% |
| Individual | – | 0.751 | 0.686 | 0.985 | 0.531 | 100% |
| Task Arithmetic | True | 93.7% (0.704) | 75.3% (0.516) | 72.6% (0.715) | **88.5%** (0.454) | 82.53% |
| TIES | True | 91.5% (0.687) | 82.1% (0.563) | 74.9% (0.738) | 86.6% (0.445) | 83.81% |
| DARE | True | **94.0%** (0.706) | 75.6% (0.518) | 72.4% (0.714) | 88.2% (0.452) | 82.56% |
| CABS | True | 67.2% (0.505) | 75.3% (0.516) | 69.9% (0.689) | 76.5% (0.392) | 72.25% |
| DOGE | False | 67.1% (0.504) | **99.5%** (0.682) | **83.7%** (0.825) | 75.5% (0.387) | 81.45% |
| OPIC(Ours) | False | 92.9% (0.698) | 81.4% (0.558) | 75.9% (0.748) | 86.6% (0.444) | **84.22%** |

lead to lower average performance compared to our method. Overall, these results indicate that OPIC generalizes well across model scales and architectures. This robustness is particularly important in practical scenarios where task distributions and model configurations may vary significantly.

### 5.3. Integrating OPIC into Existing Merging Methods.

We integrate OPIC with the representative merging methods of TIES and DARE. For more descriptions of the integration approach, please refer to Section 4.4.

Table 3 presents the normalized scores calculated via Equation (10). As demonstrated, incorporating OPIC into TIES and DARE maintains performance levels comparable to the baselines. On certain datasets such as ANLI, our approach achieves substantial improvements and even surpasses the

*Table 3.* Integration of OPIC with Baseline Methods.

| Method | ANLI(↑) | COQA(↑) | MC_TACO(↑) | MATHQA(↑) | Mean(↑) |
|---|---|---|---|---|---|
| *TIES* | | | | | |
| Base | 80.0% | 58.9% | 70.8% | 100.1% | 77.63% |
| + OPIC | 100.1% | 49.0% | 74.8% | 90.3% | 78.80% |
| *DARE* | | | | | |
| Base | 71.8% | 66.2% | 75.8% | 99.5% | 78.32% |
| + OPIC | 102.2% | 51.0% | 74.5% | 76.7% | 76.10% |

fine-tuned model, demonstrating its strong potential. Notably, our method enables existing approaches to eliminate their reliance on validation datasets and achieve automatic and efficient hyperparameter optimization.

In summary, these empirical results confirm the effective interoperability of our method with existing state-of-the-art

merging approaches. Furthermore, the adaptive parameter optimization capability of our method, which eliminates the need for test dataset access during the optimization process, addresses the data dependency bottleneck of traditional merging methods. This not only validates the rationality of the OPIC mechanism but also endows it with practical applicability in real-world scenarios where data privacy and data availability are constrained, highlighting its substantial practical value and scalability.

## 5.4. Computational Cost Analysis

Since OPIC introduces an evolutionary search procedure, we further estimate its computational cost and compare it with tuning-based baselines. We consider a typical ICL evaluation setting with $D = 20$ samples and sequence length $S = 48$, corresponding to approximately 960 tokens per evaluation. With a throughput of 150 tokens/s on a single A100 GPU, each evaluation takes about 6.4 seconds, or 0.0018 GPU hours. In our evolutionary search, the number of generations is set to $G = 20$ and the population size is $K = 15$, resulting in $E = 300$ evaluations and a total cost of approximately 0.54 GPU hours. For comparison, we include additional baseline methods under both one-shot and grid-search settings. Since the test set contains more samples and longer texts, we assume that each evaluation round for the other algorithms takes 0.01 hours, and set the number of grid-search trials to 20.

*Table 4.* Estimated computational cost of different merging settings.

| Method | Eval | Cost/Eval (h) | Total (GPU h) | Remarks |
|---|---|---|---|---|
| TA / DARE / TIES | 1 | ∼0 | ∼0 | One-shot merging |
| TA / DARE / TIES | 20 | 0.01 | ∼0.2 | Grid-search tuning |
| OPIC (Ours) | 300 | 0.0018 | ∼0.54 | Evolutionary optimization |

As shown in Table 4, in the commonly used tuning-based setting, OPIC incurs about 2–3× the GPU cost of grid-search baselines. This additional cost is acceptable in practice because OPIC removes the requirement for task-specific labeled validation data and provides a more automated merging strategy.

## 5.5. Ablation Studies

**Generalization and robustness evaluation.** To further assess the generalization and robustness of our approach, we also conducted experiments on datasets outside the fine-tuning test set and compared our approach with mainstream model merging methods.

As shown in Table 5, we evaluate reasoning, bias, and comprehension and generation using ARC, CROWS-PAIRS, and MMLU, respectively. For ARC and MMLU, higher scores denote better performance, whereas for CROWS_PAIRS, a score closer to 0.5 indicates lower bias. The results show

*Table 5.* Out-of-Domain Performance of the Fusion Models.

| Method | ARC(↑) | CROWS_PAIRS(=) | MMLU(↑) |
|---|---|---|---|
| Task Arithmetic | 0.645 | 0.587 | 0.332 |
| TIES | **0.761** | 0.614 | 0.533 |
| DARE | 0.75 | 0.615 | 0.528 |
| CABS | 0.754 | 0.634 | **0.589** |
| DOGE | 0.617 | 0.577 | 0.300 |
| OPIC(Ours) | 0.653 | **0.570** | 0.544 |

that our method achieves the best performance in mitigating model bias and excels in comprehension tasks. However, it performs slightly worse than methods such as TIES and DARE on reasoning tasks, which may be because these approaches adopt pruning-based task vector modification strategies that are more conducive to improving generalization in reasoning. In comprehension and generation tasks, our method also achieves leading performance.

**Effects of each module.** Our method influences the model merging process through refinement operators consisting of three hierarchical refinement matrices (A, B, C). To evaluate the contribution of each component, we start from the full OPIC algorithm and progressively restrict one matrix at a time, reporting normalized performance scores. The results are summarized in Table 6. Removing the inter-layer refinement (matrix A) and the task-vector–level refinement (matrix B) leads to average performance drops of 2.9% and 2.4%, respectively. Further restricting both matrices B and C results in a larger degradation of 4.3%. These results demonstrate the effectiveness and complementarity of the proposed components, and also indicate that applying multi-dimensional and hierarchical refinements to task vectors is beneficial for mitigating task conflict.

*Table 6.* Component analysis of the OPIC framework. We report the performance impact of removing specific modules.

| Configuration | ANLI(↑) | COQA(↑) | MC_TACO(↑) | MATHQA(↑) | Mean(↑) |
|---|---|---|---|---|---|
| Pre-trained | 65.3% | 66.6% | 65.8% | 82.7% | 70.10% |
| OPIC (Ours) | 97.5% | 65.0% | 77.4% | 83.0% | 80.70% |
| w/o A | 97.3% | 60.5% | 75.7% | 77.7% | 77.80% |
| w/o B | 95.1% | 60.8% | 76.8% | 80.5% | 78.30% |
| w/o (A & B) | 93.9% | 60.1% | 74.2% | 77.2% | 76.40% |

**Hyperparameter sensitivity.** The primary hyperparameter in our method is the search range of the refinement operators. We progressively relax this range from [0, 1] to [0, 1.2] and [0, 1.5]; the corresponding performance under different settings is reported in Figure 4. The results indicate that widening the search range has no significant impact on performance, with the maximum difference being 0.36%. This consistent performance across different ranges demonstrates the robustness of our method. This insensitivity to the search range is practically significant, as

it demonstrates the robustness of OPIC and suggests that we can achieve optimal performance without the need for meticulous hyperparameter tuning.

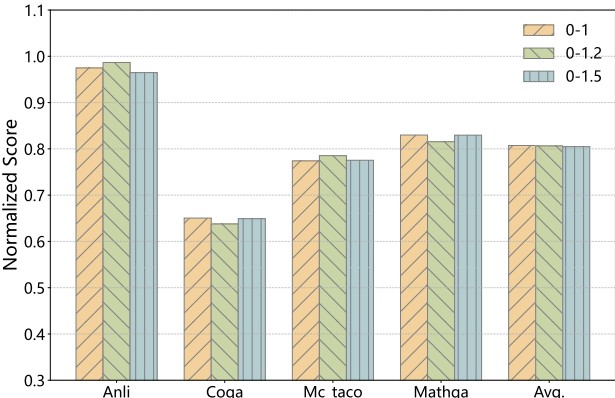

*Figure 4.* Sensitivity analysis of the search range.

## 6. Conclusion and Discussion

**Conclusion.** This work revisits performance degradation in model merging from a new perspective and identifies the loss of ICL (ICL) capability as the key indicator, going beyond conventional parameter-level explanations. Building on this observation, we propose OPIC, a validation-free model merging framework that combines hierarchical merging operators with evolutionary optimization on self-generated data. Extensive experiments demonstrate that OPIC consistently outperforms existing approaches across model scales and architectures, while remaining robust in cross-domain settings. Moreover, we provide a practical pathway for seamlessly integrating OPIC with existing model merging methods, which preserves their performance while eliminating the reliance on validation data and improving practicality. We believe this work opens new directions for scalable and privacy-preserving model merging.

**Discussion.** Although OPIC demonstrates strong performance on text-based language models, our evaluation is currently restricted to unimodal settings. Extending the framework to multimodal foundation models presents additional challenges, including the effective curation and utilization of multimodal data, as well as the appropriate characterization of in-context capability in multimodal architectures. Moreover, while OPIC has been evaluated across diverse model architectures and scales, its scalability to larger models and the fusion of a greater number remains insufficiently explored. Addressing these challenges represents an important direction for future research.

## Impact Statement

This paper presents work whose goal is to advance the field of Machine Learning. There are many potential societal consequences of our work, none which we feel must be specifically highlighted here."

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

## A. Main hypotheses and representative methods of data-free merging

*Table 7.* Main Hypotheses and Descriptions of model merging

| Hypothesis | Approach | Explanation |
|---|---|---|
| Redundancy | Eliminate redundancy by retaining, according to specific rules, the most influential components of each task vector ($\tau$) and setting the remaining components to zero. | During fine-tuning, many model parameters may change but have minimal impact on performance. However, when merging parameters that are influential for one model but redundant (i.e., non-influential) for others, the influential values can be overshadowed by redundant ones, thereby degrading overall model performance(Yadav et al., 2023; Yu et al., 2024; Du et al., 2024; Yang et al., 2025). |
| Directional Conflict | Resolve sign conflicts between models by determining the aggregated sign for each parameter, taking into account the overall direction across all task vectors. | A given parameter may have a positive value for some task vectors and a negative value for others. Simply averaging these values can thus impair performance on both tasks(Yadav et al., 2023; Yang et al., 2023). |

## B. REGBench

REGBench is generated via customized finite-state machines and random sampling, offering controllable complexity and well-defined probability distributions. This makes it particularly suitable for assessing model performance on in-context language learning tasks.

Let the test set contain $n$ problem instances. Each instance, $d^{(i)}$, is a sequence of strings, $[d_1^{(i)}, d_2^{(i)}, \ldots, d_m^{(i)}]$, where all strings are sampled from the same unknown probability language $L^{(i)}$. The model's objective is to predict the next token, $x_j$, given the context $d_{<j}^{(i)}$ (i.e., all characters preceding the $j$-th position in instance $d^{(i)}$).

- Step 1: model merging: Based on a candidate solution, i.e., the parameter combination $(A, B, C)$, modify the task vectors, and the fused model $\theta_{\text{fused}}$ is constructed using Equation 8.

- Step 2: Fitness Evaluation: The model's performance is evaluated on the REGBench validation set, and its accuracy is calculated according to Equation X to serve as the fitness function. The objective is to maximize this fitness value. Individuals with higher fitness have a greater probability of being selected and passing their genetic material (i.e., superior parameters) to the next generation.

- Step 3: Crossover and Mutation: 1.Crossover:Selected individuals are paired up to exchange parts of their chromosomes. This helps combine advantageous genes into the offspring while maintaining population diversity. 2.Mutation:Random mutations are introduced into the offspring's chromosomes with a small probability. In this paper, this is achieved by adding Gaussian noise to a random element in the $A$, $B$, or $C$ matrices. Mutation introduces new genetic material into the population, preventing the algorithm from prematurely converging to a local optimum.

- Repeat the above steps until a termination condition is met.

## C. Experiment Details

### C.1. Evaluation Metrics

To provide a consistent and comparable basis for evaluating model performance across various benchmarks., the following metrics were used:

- Anli: Accuracy

- Coqa: Exact Match

- Mc_taco: Accuracy

- Mathqa: Accuracy

- Arc: Accuracy

- Crows_pairs: pct_stereotype

- Mmlu: Accuracy

## C.2. Compared Baselines

- Pre-trained: Uses a pre-trained model for each task without integrating task-specific information. Serves as a basic benchmark for comparison.

- Individual: Fine-tunes a separate model for each task, ensuring no task interference and providing an ideal baseline for task-specific performance.

- Task Arithmetic: Computes task vectors for individual tasks and sums them up to construct a multi-task vector. This vector is scaled by a coefficient ($\lambda$) and added to the pre-trained model's parameters.

- Ties-Merging: Combines steps like trimming, parameter sign determination, and disjoint merging to produce a merged task vector $\tau$. The final model is defined as $\theta = \theta_0 + \lambda_{tau}$, where $\lambda_{tau}$ is tuned using a validation set.

- CABS: By employing masking and pruning strategies, the CABS method alleviates the issues of high parameter overlap and skewed weight distributions during model merging, effectively resolving conflicts between task vectors.

- DOGE: The DOGE frames model merging as a constrained optimization problem (i.e., minimizing the gap between the merged model and individual models, subject to the constraint of retaining shared knowledge) and solve it via adaptive projective gradient descent.

## C.3. Datasets and Models

*Table 8.* Introduction of Datasets

| Task Family | Description |
| --- | --- |
| Anli(Nie et al., 2020) | Adversarial natural language inference tasks designed to test model robustness. |
| Coqa(Reddy et al., 2019) | Conversational question answering tasks to test dialog understanding. |
| Mc_taco(Zhou et al., 2019) | Question-answer pairs that require temporal commonsense comprehension. |
| Mathqa(Amini et al., 2019) | Question answering tasks involving mathematical reasoning and problem-solving. |
| Arc(Clark et al., 2018) | Tasks involving complex reasoning over a diverse set of questions. |
| Crows_pairs(Nangia et al., 2020) | Tasks designed to test model biases in various sociodemographic groups. |
| Mmlu(Hendrycks et al., 2021) | Massive Multitask Language Understanding benchmark for broad domain language evaluation. Several variants are supported. |

*Table 9.* Introduction of Models

| Models | Description |
|---|---|
| Qwen2.5(Team, 2024) | Qwen 2.5 is released by Alibaba Cloud in September 2024. Regarded as a comprehensive upgrade to Qwen 2, this iteration achieves substantial breakthroughs primarily in instruction following, mathematics, coding, and logical reasoning. |
| Llama3.2(AI@Meta, 2024) | Released by Meta in September 2024, Llama 3.2 marks the inaugural integration of multimodal (vision-language) capabilities within the Llama family, while also significantly expanding the applicability of lightweight models. |

## C.4. Grid Search Details

Following the prior works, algorithms such as TIES, DARE, CABS optimize their hyperparameter $\lambda$ using grid search, we set the search range to [0.1, 1] with a step size of 0.1. For TIES-Merging, which require a masking ratio, we set the the search range to [0.1, 0.3] with a step size of 0.1. For DARE, which require a drop ratio, we set the search range to [0.1, 0.9] with a step size of 0.2. For algorithms that use evolutionary strategies for coefficient search, we set the range to [0.1, 1] to facilitate continuous variable search.

## C.5. Training details

All large language models were trained for 1 epoch using a learning rate of 0.0001. Training was conducted in bfloat16 with the Adafactor optimizer to reduce GPU memory usage. The maximum sequence length was 256 tokens, and the batch size was set to 1. Neither a learning rate scheduler nor weight decay was employed in any training procedure.

## C.6. Hardware and Evaluation Platform

Model fine-tuning is performed on A100 GPUs with 80GB of RAM. Model merging and evaluation was conducted using lm-evaluation-harness(Gao et al., 2024) on an NVIDIA RTX 4090 GPU with 24GB of RAM, and the observed result variance was small and stable.

## C.7. Detail Results

The following table details the raw scores of each method under different parameter settings.

*Table 10.* Detailed results of Task Arithmetic under different parameter settings

| Task Arithmetic | 0.1 | 0.2 | 0.3 | 0.4 | 0.5 | 0.6 | 0.7 | 0.8 | 0.9 | 1.0 |
|---|---|---|---|---|---|---|---|---|---|---|
| Qwen2.5-1.5B | | | | | | | | | | |
| Anli | 0.524 | 0.597 | 0.621 | 0.621 | 0.611 | 0.585 | 0.556 | 0.520 | 0.499 | 0.466 |
| Coqa | 0.359 | 0.358 | 0.403 | 0.367 | 0.313 | 0.244 | 0.161 | 0.132 | 0.112 | 0.090 |
| Mc_taco | 0.684 | 0.695 | 0.714 | 0.724 | 0.734 | 0.730 | 0.707 | 0.697 | 0.691 | 0.686 |
| Mathqa | 0.415 | 0.371 | 0.340 | 0.308 | 0.281 | 0.268 | 0.250 | 0.238 | 0.227 | 0.218 |
| Qwen2.5-3B | | | | | | | | | | |
| Anli | 0.509 | 0.497 | 0.493 | 0.486 | 0.470 | 0.437 | 0.390 | 0.356 | 0.338 | 0.333 |
| Coqa | 0.644 | 0.585 | 0.525 | 0.458 | 0.426 | 0.374 | 0.294 | 0.215 | 0.125 | 0.059 |
| Mc_taco | 0.784 | 0.781 | 0.809 | 0.849 | 0.847 | 0.752 | 0.619 | 0.492 | 0.404 | 0.355 |
| Mathqa | 0.473 | 0.485 | 0.494 | 0.497 | 0.492 | 0.488 | 0.474 | 0.464 | 0.455 | 0.444 |
| Llama3.2-3B | | | | | | | | | | |
| Anli | 0.342 | 0.469 | 0.596 | 0.661 | 0.704 | 0.722 | 0.730 | 0.727 | 0.724 | 0.726 |
| Coqa | 0.651 | 0.637 | 0.600 | 0.569 | 0.516 | 0.429 | 0.328 | 0.248 | 0.146 | 0.100 |
| Mc_taco | 0.466 | 0.585 | 0.720 | 0.739 | 0.715 | 0.694 | 0.684 | 0.679 | 0.678 | 0.675 |
| Mathqa | 0.375 | 0.402 | 0.428 | 0.445 | 0.454 | 0.457 | 0.461 | 0.457 | 0.450 | 0.436 |

*Table 11.* Detailed results of CABS under different parameter settings

| CABS | 0.1 | 0.2 | 0.3 | 0.4 | 0.5 | 0.6 | 0.7 | 0.8 | 0.9 | 1.0 |
|---|---|---|---|---|---|---|---|---|---|---|
| Qwen2.5-1.5B | | | | | | | | | | |
| Anli | 0.432 | 0.469 | 0.482 | 0.483 | 0.486 | 0.475 | 0.481 | 0.469 | 0.465 | 0.461 |
| Coqa | 0.369 | 0.140 | 0.170 | 0.168 | 0.156 | 0.145 | 0.152 | 0.137 | 0.128 | 0.117 |
| Mc_taco | 0.698 | 0.680 | 0.684 | 0.669 | 0.667 | 0.666 | 0.666 | 0.665 | 0.664 | 0.664 |
| Mathqa | 0.394 | 0.408 | 0.401 | 0.391 | 0.379 | 0.359 | 0.344 | 0.333 | 0.319 | 0.306 |
| Qwen2.5-3B | | | | | | | | | | |
| Anli | 0.490 | 0.510 | 0.524 | 0.520 | 0.513 | 0.513 | 0.504 | 0.502 | 0.500 | 0.497 |
| Coqa | 0.657 | 0.638 | 0.609 | 0.592 | 0.567 | 0.532 | 0.510 | 0.462 | 0.445 | 0.421 |
| Mc_taco | 0.743 | 0.775 | 0.769 | 0.759 | 0.752 | 0.745 | 0.717 | 0.706 | 0.711 | 0.719 |
| Mathqa | 0.424 | 0.449 | 0.464 | 0.472 | 0.477 | 0.478 | 0.480 | 0.476 | 0.477 | 0.476 |
| Llama3.2-3B | | | | | | | | | | |
| Anli | 0.333 | 0.334 | 0.347 | 0.357 | 0.385 | 0.407 | 0.437 | 0.474 | 0.505 | 0.520 |
| Coqa | 0.658 | 0.654 | 0.642 | 0.623 | 0.603 | 0.589 | 0.569 | 0.538 | 0.516 | 0.459 |
| Mc_taco | 0.573 | 0.682 | 0.711 | 0.705 | 0.696 | 0.690 | 0.687 | 0.688 | 0.689 | 0.693 |
| Mathqa | 0.355 | 0.364 | 0.370 | 0.372 | 0.375 | 0.374 | 0.382 | 0.385 | 0.392 | 0.396 |

*Table 12.* Detailed results of DARE under different parameter settings

| DARE | 0.1_0.3 | 0.1_0.5 | 0.1_0.7 | 0.1_0.9 | 0.3_0.3 | 0.3_0.5 | 0.3_0.7 | 0.3_0.9 | 0.5_0.3 | 0.5_0.5 |
|---|---|---|---|---|---|---|---|---|---|---|
| Qwen2.5-1.5B | | | | | | | | | | |
| Anli | 0.460 | 0.455 | 0.360 | 0.331 | 0.603 | 0.633 | 0.563 | 0.457 | 0.611 | 0.576 |
| Coqa | 0.384 | 0.417 | 0.383 | 0.440 | 0.359 | 0.377 | 0.382 | 0.104 | 0.231 | 0.239 |
| Mc_taco | 0.715 | 0.723 | 0.680 | 0.664 | 0.729 | 0.670 | 0.691 | 0.724 | 0.774 | 0.677 |
| Mathqa | 0.411 | 0.415 | 0.408 | 0.409 | 0.338 | 0.328 | 0.325 | 0.301 | 0.269 | 0.274 |
| Qwen2.5-3B | | | | | | | | | | |
| Anli | 0.496 | 0.498 | 0.509 | 0.476 | 0.479 | 0.461 | 0.493 | 0.463 | 0.379 | 0.376 |
| Coqa | 0.648 | 0.650 | 0.642 | 0.649 | 0.538 | 0.546 | 0.547 | 0.497 | 0.359 | 0.458 |
| Mc_taco | 0.785 | 0.769 | 0.789 | 0.807 | 0.833 | 0.762 | 0.716 | 0.810 | 0.775 | 0.772 |
| Mathqa | 0.480 | 0.482 | 0.471 | 0.471 | 0.498 | 0.499 | 0.499 | 0.485 | 0.493 | 0.493 |
| Llama3.2-3B | | | | | | | | | | |
| Anli | 0.349 | 0.345 | 0.345 | 0.345 | 0.624 | 0.616 | 0.611 | 0.582 | 0.721 | 0.712 |
| Coqa | 0.651 | 0.653 | 0.660 | 0.652 | 0.591 | 0.600 | 0.588 | 0.632 | 0.501 | 0.504 |
| Mc_taco | 0.455 | 0.452 | 0.452 | 0.452 | 0.716 | 0.705 | 0.712 | 0.705 | 0.714 | 0.722 |
| Mathqa | 0.377 | 0.376 | 0.378 | 0.379 | 0.427 | 0.426 | 0.430 | 0.424 | 0.448 | 0.449 |

| TIES | 0.5_0.7 | 0.5_0.9 | 0.7_0.3 | 0.7_0.5 | 0.7_0.7 | 0.7_0.9 | 0.9_0.3 | 0.9_0.5 | 0.9_0.7 | 0.9_0.9 |
|---|---|---|---|---|---|---|---|---|---|---|
| Qwen2.5-1.5B | | | | | | | | | | |
| Anli | 0.609 | 0.435 | 0.525 | 0.574 | 0.455 | 0.347 | 0.511 | 0.435 | 0.353 | 0.329 |
| Coqa | 0.273 | 0.127 | 0.118 | 0.110 | 0.105 | 0.000 | 0.108 | 0.073 | 0.000 | 0.000 |
| Mc_taco | 0.742 | 0.663 | 0.737 | 0.708 | 0.669 | 0.629 | 0.668 | 0.667 | 0.661 | 0.565 |
| Mathqa | 0.267 | 0.247 | 0.241 | 0.246 | 0.223 | 0.195 | 0.213 | 0.225 | 0.211 | 0.185 |
| Qwen2.5-3B | | | | | | | | | | |
| Anli | 0.467 | 0.376 | 0.354 | 0.364 | 0.334 | 0.319 | 0.339 | 0.333 | 0.332 | 0.330 |
| Coqa | 0.360 | 0.348 | 0.189 | 0.181 | 0.233 | 0.032 | 0.037 | 0.045 | 0.097 | 0.000 |
| Mc_taco | 0.698 | 0.661 | 0.440 | 0.405 | 0.762 | 0.664 | 0.341 | 0.375 | 0.381 | 0.576 |
| Mathqa | 0.493 | 0.465 | 0.468 | 0.469 | 0.477 | 0.432 | 0.449 | 0.444 | 0.431 | 0.213 |
| Llama3.2-3B | | | | | | | | | | |
| Anli | 0.702 | 0.706 | 0.733 | 0.738 | 0.734 | 0.722 | 0.737 | 0.736 | 0.726 | 0.719 |
| Coqa | 0.492 | 0.518 | 0.314 | 0.328 | 0.298 | 0.246 | 0.135 | 0.196 | 0.154 | 0.191 |
| Mc_taco | 0.720 | 0.714 | 0.695 | 0.697 | 0.690 | 0.683 | 0.677 | 0.677 | 0.661 | 0.673 |
| Mathqa | 0.451 | 0.452 | 0.447 | 0.456 | 0.449 | 0.444 | 0.430 | 0.433 | 0.436 | 0.419 |

*Table 13.* Detailed results of TIES under different parameter settings

| TIES | 0.1_0.1 | 0.1_0.2 | 0.1_0.3 | 0.3_0.1 | 0.3_0.2 | 0.3_0.3 | 0.5_0.1 | 0.5_0.2 |
|---|---|---|---|---|---|---|---|---|
| Qwen2.5-1.5B | | | | | | | | |
| Anli | 0.461 | 0.474 | 0.507 | 0.531 | 0.607 | 0.641 | 0.557 | 0.629 |
| Coqa | 0.422 | 0.389 | 0.371 | 0.126 | 0.229 | 0.296 | 0.199 | 0.279 |
| Mc_taco | 0.681 | 0.672 | 0.675 | 0.662 | 0.663 | 0.671 | 0.673 | 0.672 |
| Mathqa | 0.408 | 0.416 | 0.421 | 0.398 | 0.373 | 0.360 | 0.354 | 0.326 |
| Qwen2.5-3B | | | | | | | | |
| Anli | 0.506 | 0.508 | 0.497 | 0.501 | 0.492 | 0.493 | 0.489 | 0.484 |
| Coqa | 0.647 | 0.653 | 0.651 | 0.561 | 0.548 | 0.544 | 0.450 | 0.451 |
| Mc_taco | 0.691 | 0.714 | 0.735 | 0.695 | 0.718 | 0.748 | 0.687 | 0.748 |
| Mathqa | 0.436 | 0.455 | 0.467 | 0.485 | 0.485 | 0.488 | 0.488 | 0.500 |
| Llama3.2-3B | | | | | | | | |
| Anli | 0.333 | 0.328 | 0.334 | 0.364 | 0.469 | 0.545 | 0.482 | 0.616 |
| Coqa | 0.659 | 0.656 | 0.654 | 0.656 | 0.644 | 0.635 | 0.647 | 0.616 |
| Mc_taco | 0.424 | 0.430 | 0.439 | 0.457 | 0.528 | 0.590 | 0.546 | 0.702 |
| Mathqa | 0.359 | 0.363 | 0.371 | 0.383 | 0.395 | 0.407 | 0.403 | 0.429 |

| TIES | 0.5_0.3 | 0.7_0.7 | 0.7_0.2 | 0.7_0.3 | 0.9_0.1 | 0.9_0.2 | 0.9_0.3 | |
|---|---|---|---|---|---|---|---|---|
| Qwen2.5-1.5B | | | | | | | | |
| Anli | 0.656 | 0.581 | 0.607 | 0.615 | 0.562 | 0.589 | 0.575 | |
| Coqa | 0.316 | 0.178 | 0.241 | 0.220 | 0.133 | 0.163 | 0.144 | |
| Mc_taco | 0.688 | 0.680 | 0.686 | 0.704 | 0.690 | 0.698 | 0.709 | |
| Mathqa | 0.313 | 0.325 | 0.292 | 0.281 | 0.298 | 0.266 | 0.253 | |
| Qwen2.5-3B | | | | | | | | |
| Anli | 0.476 | 0.475 | 0.467 | 0.442 | 0.469 | 0.437 | 0.400 | |
| Coqa | 0.437 | 0.389 | 0.401 | 0.379 | 0.357 | 0.357 | 0.260 | |
| Mc_taco | 0.802 | 0.681 | 0.793 | 0.814 | 0.693 | 0.832 | 0.605 | |
| Mathqa | 0.505 | 0.489 | 0.504 | 0.499 | 0.498 | 0.495 | 0.480 | |
| Llama3.2-3B | | | | | | | | |
| Anli | 0.679 | 0.578 | 0.687 | 0.721 | 0.646 | 0.713 | 0.731 | |
| Coqa | 0.585 | 0.623 | 0.563 | 0.506 | 0.600 | 0.493 | 0.383 | |
| Mc_taco | 0.743 | 0.667 | 0.738 | 0.718 | 0.733 | 0.713 | 0.687 | |
| Mathqa | 0.430 | 0.420 | 0.445 | 0.451 | 0.433 | 0.458 | 0.455 | |

