# OpenReview forum: "OPIC: Enhancing Language Model Merging via Optimizing In-Context Capability"
_ICML.cc/2026/Conference — ICML 2026 regular_

### Official Review · Reviewer_dYhK · 2026-02-14

**Soundness:** 3
**Presentation:** 3
**Significance:** 3
**Originality:** 3
**Overall Recommendation:** 5
**Confidence:** 3

**Summary:**

This paper proposed OPIC, a task-vector based merging framework that treats merging as an optimization problem - where the objective is to preserve ICL capability (as measured on REGBench). OPIC find a hierarchical refinement operator using a genetic algorithm, bypassing the traditional dependence of hyperparameter tuning on the validation set. The method shows improved retention of downstream tasks.

**Compliance With Llm Reviewing Policy:**

Affirmed.

**Final Justification:**

The authors sufficiently addressed by concerns in the rebuttal. I am revising my score under the assumption that the discussed changes will be incorporated into the paper.

**Key Questions For Authors:**

1. What evidence supports the claim that ICL loss causes task conflict, rather than being a side effect of merges that are too aggressive? Can you think of a more controlled test that separates these possibilities?

2. What is the compute cost of OPIC search (GPU-hours), and how sensitive are results to search budget and random seed?

**Limitations:**

No. The authors discuss some limitations in Section 6 (Under Discussion), but it would be better to label these as Limitations.

I would also add the following to the Impact Statement, which is currently very brief:
- Even if OPIC itself is not “directly harmful,” merging methods can lower barriers to distributing modified models.
- “Validation-free” merging can help in privacy-constrained settings, but it should be paired with appropriate evaluation to avoid silent regressions.

**Strengths And Weaknesses:**

**Strengths:**

1. Clear formulation - The A,B,C weights provide an inspectable control surface, and the paper explains how these compose with existing vector-modification methods like TIES. This means you can treat OPIC as a meta-tuner that sits on top of merging pipelines; rather than a monolithic new merging rule.

2. The preliminary experiment probes for a clear correlation between REGBench ICL accuracy and downstream retention across $\lambda$, using this as motivation for the optimization objective. Even if the causal story is not fully proven, the paper does a better job than many merging papers at grounding the objective in an observable behavioral metric rather than only empirical observations.

3. Empirical improvements on various models and tasks.

**Weaknesses:**

1. The paper revolves around the core claim that ICL loss is a large cause of task conflict. However, evidence towards this is insufficient. The experiment showing correlation along $\lambda$ along 1 backbone and task does not cleanly separate ICL from REGBench accuracy for avoiding destructive updates. A more convincing causal test would manipulate ICL independently of task vector interference (e.g., keep ICL fixed while varying conflict, or vice versa).

2. OPIC optimizes only one proxy metric (REGBench ICL) but is evaluated on several downstream tasks. It is unclear whether preserving REGBench ICL is reliably aligned with preserving performance on all target tasks; the mixed results on out-of-domain reasoning tasks suggest the proxy may not always match the goal.

3. Since OPIC is a search procedure, search cost and budget are crucial components to report. However, these have not been clearly reported.

---

> ### Author Rebuttal · Authors · 2026-03-30
>
> Thank you for recognizing the contributions of our work—especially the advantage that our method can be **combined with other approaches**, and the **scientific rigor** of our empirical strategy. Your questions and suggestions are very helpful. Below we respond to your concerns about **causal evidence**, **generalization**, and **computational cost**.
>
> ### **W1 & Q1. Lack of Causal Evidence**
>
> Thank you for the constructive criticism. We agree that our current empirical evidence primarily establishes a positive correlation between ICL capability and downstream performance degradation during merging, rather than a strict causal relationship. Based on other reviewers' suggestions and careful reflection, we believe it would be better to revise the causal statement to correlation.
>
> You also suggest conducting more rigorous experiments to distinguish the performance impact of conflicts between ICL and merging itself. We wholeheartedly agree that. Therefore, we conducted the following controlled experiments, using prompt words to fine-tune the ICL ability of the repaired fusion model without affecting the parameters of the fusion model. Specifically:
>
> * Merge four tasks (ANLI, MCTACO, COQA, MATHQA) using **Task Arithmetic** on **Qwen1.5-7B**.
> * Apply **Soft Prompt Tuning** on the merged model to improve ICL capability **without changing model parameters or task knowledge**.
> * Evaluate on **REGBench** and downstream tasks.
>
> **Obeservations:**
> - The merged model shows a **clear degradation in ICL capability** (REGBench) compared to the pretrained model, **together with** a drop in downstream performance.
> - After improving ICL capability via prompt tuning, the model achieves **consistent gains on REGBench** and **noticeable improvements** across downstream tasks.
>
> **Conclusion:**
> Overall, these results suggest that **manipulating ICL capability** (while controlling for task knowledge and model parameters) leads to **corresponding changes** in downstream performance, providing **stronger empirical support** for a causal connection between ICL capability and task conflict. We will include this experiment in the final version.
>
> **Table 1. Intervention study on ICL capability and downstream performance.**
>
> | Model | REGBench (↑) | ANLI (↑) | MCTACO (↑) | COQA (↑) | MATHQA (↑) |
> |---|---|---|---|---|---|
> | Pretrained | 0.864 | 0.414 | 0.627 | 0.420 | 0.345 |
> | Individual | — | 0.634 | 0.953 | 0.630 | 0.417 |
> | Merged (TA) | 0.726 | 0.611 | 0.734 | 0.313 | 0.281 |
> | Merged + Prompt Tuning | 0.785 | 0.655 | 0.702 | 0.371 | 0.294
>
> ---
>
> ### **W2. Domain Generalization**
>
> Thank you for raising this important point. You are concerned about whether **REGBench**, as a proxy for ICL capability, generalizes across diverse downstream tasks.
>
> We view this as a technical advantage of our approach, and we agree that the current draft could explain REGBench more clearly and discuss why it can work as a general proxy. Our key points are:
>
> - **How REGBench is generated.** REGBench is created by an automatic generator and consists of **structured, regular strings**. It does **not** inherently encode any domain-specific knowledge.
> - **Why it can generalize.** REGBench evaluates the model’s **ICL capability**, and our method leverages the empirically observed relationship between **ICL capability** and **task conflict** to predict downstream performance. Since stronger ICL capability broadly benefits many downstream tasks, using REGBench as a proxy can generalize across domains.
>
> ---
>
> ### **W3 & Q2. Computational Cost**
>
> We thank the reviewer for pointing out the missing computational discussion. We will add a dedicated paragraph reporting the compute trade-off and running time in the final version. Below we give a transparent back-of-the-envelope estimate of OPIC in GPU-hours, using a 3B parameter model on a single A100 GPU.
>
> - Typical ICL benchmark: **D = 20**, **S = 48** → ~**960 tokens** per evaluation.
> - Throughput assumption: **150 tokens/s** → ~**6.4 s/eval** → **0.0018 GPU-hours/eval**.
> - Evolutionary search: **G = 20**, **K = 15** → **E = 300 evals** → total **~0.54 GPU-hours**.
>
> Baselines (TA / DARE / TIES) are one-shot for a single merge, but typically require tuning. With **E = 20** candidates and **0.01 GPU-hours/eval** on a validation set, total cost is **~0.2 GPU-hours**.
>
> **Table 2. Estimated computational cost.**
>
> | Method | #Eval (E) | Cost/Eval (h) | Total (GPU h) | Remarks |
> |---|---:|---:|---:|---|
> | TA / DARE / TIES | 1 | ~0 | ~0 | One-shot merging |
> | TA / DARE / TIES | 20 | 0.01 | ~0.2 | Grid-search tuning |
> | **OPIC (Ours)** | **300** | **0.0018** | **~0.54** | Evolutionary optimization |
>
> **Conclusion:** Overall, the overall GPU time is about **2–3×** tuning-based baselines.—while avoiding labeled validation data and enabling a more automated, general merging procedure. We will submit a detailed analysis and calculation of the calculated costs in the final version

---

> > ### Author Rebuttal · Reviewer_dYhK · 2026-04-03
> >
> > Thank you for the clarifications. Based on the new information, I have revised my score. (I am revising my score under the assumption that the discussed changes will be incorporated into the paper.)

---

> > > ### Author Response · Authors · 2026-04-05
> > >
> > > Thank you for your follow-up and for revising your score based on the rebuttal. We appreciate your recognition that the clarifications were helpful. As you noted, the score revision is based on the assumption that the discussed changes will be incorporated into the paper, and we confirm that these revisions will be reflected in the final version.

---

### Official Review · Reviewer_5PXc · 2026-03-02

**Soundness:** 3
**Presentation:** 3
**Significance:** 3
**Originality:** 3
**Overall Recommendation:** 5
**Confidence:** 4

**Summary:**

This paper proposes a validation-free model merging framework that aims to improve merging performance by optimizing in-context learning (ICL) capability. Through preliminary experiments, the authors demonstrate a strong correlation between ICL degradation and performance decline during model merging, suggesting that the loss of ICL ability is a key factor underlying task conflict. Motivated by this observation, the merging process is formulated as an optimization problem, where preserving ICL serves as the objective, and a genetic algorithm is employed to search for optimal refinement parameters. In addition, the method leverages a self-generated dataset for evaluation, thereby eliminating the dependence on validation sets that most existing merging approaches require.

Overall, this work offers an interesting and novel perspective on model merging by focusing on preserving ICL capability. The proposed validation-free merging framework is innovative and practically valuable. Experimental results show that the proposed method is practical and achieves competitive.

**Compliance With Llm Reviewing Policy:**

Affirmed.

**Final Justification:**

The authors' rebuttal has addressed all of my concerns. On the causality issue, the additional controlled experiment alleviates my doubts about the credibility of the paper's core hypothesis. It is appropriate that the authors revise the original strong causal claim to a correlation-based interpretation. Overall, I acknowledge the paper's key findings and methodological innovation, and believe it meaningfully improves the practicality and usability of model merging approaches. Therefore, I am inclined to raise my score to 5.

**Key Questions For Authors:**

1. Causal Relationship Between ICL and Task PerformanceICL
Your central claim suggests that ICL degradation is a key cause of task conflict. Could you provide additional evidence (e.g., controlled ablations or intervention-style experiments) to support a causal relationship rather than correlation?

2. Computational Efficiency
What is the computational overhead of the genetic algorithm compared to grid search or gradient-based approaches? Please provide runtime, number of evaluations, and GPU cost comparisons.

3. Scalability to More Tasks
Have you tested OPIC in scenarios involving more than four tasks (e.g., 8–16 tasks)? If not, how do you expect performance and optimization stability to scale?

**Limitations:**

yes

**Strengths And Weaknesses:**

**Strengths**

1. Novel Perspective on Task Conflict

The paper proposes that degradation of ICL capability is a primary indicator of task conflict in model merging. This represents an original and insightful reframing of the problem beyond traditional parameter-level explanations and may open new research directions in understanding task conflict..

2. Practical Contribution

The proposed validation-free model merging framework is highly valuable in real-world scenarios, particularly under privacy constraints or deployment settings where validation data is unavailable. Moreover, the method integrates seamlessly with existing approaches such as TIES and DARE, extending their usability by eliminating their reliance on validation datasets .

3. Methodological Innovation

The introduction of an optimization-based framework, particularly the use of genetic algorithms for automated parameter search in model merging, is technically meaningful and improves automation compared to heuristic tuning.

**Weaknesses**

1. Limited Causal Analysis

The claim that ICL degradation is the key cause of task conflict is primarily supported by correlation analysis. The paper lacks deeper causal investigation or theoretical justification to firmly establish this claim.

2. Computational Cost of Genetic Algorithms

Applying genetic algorithms to large language models may introduce non-trivial computational overhead. The paper does not provide time or memory cost comparisons with existing methods.

3. Limited Task Diversity

Experiments involve merging only four fine-tuning tasks. The scalability of the method to larger multi-task settings (e.g., merging 8–16 tasks) is not explored, leaving open questions about broader applicability.

4.Overly Strong Causal Wording

Claims made in parts of the article (e.g., "contextual competency is the main cause of task conflicts") may exaggerate empirical evidence. More careful wording, such as key factors, is more rigorous.

---

> ### Author Rebuttal · Authors · 2026-03-30
>
> Thank you for recognizing the contributions of our work—especially (i) the **new analytical perspective on task conflict**, (ii) the **practical advantages for deployment**, and (iii) the **innovation in automation**. Your questions and suggestions are very helpful. Below we respond to your concerns.
>
> ### **W1 & Q1. Limited Causal Analysis**
>
> Thank you for pointing out this important issue on limited causal analysis. Based on the reviewers' suggestions and careful reflection, we believe it would be better to revise the causal statement to correlation. In order to further support the relationship between ICL and Task Performance, we also add following **controlled experiments**, which will be corrected in the final version of the manuscript.
>
> * Merge four tasks (ANLI, MCTACO, COQA, MATHQA) using **Task Arithmetic** on **Qwen1.5-7B**.
> * Apply **Soft Prompt Tuning** on the merged model with **DROID** (unrelated to merging tasks) to improve ICL capability **without changing model parameters or task knowledge**.
> * Evaluate on **REGBench** and downstream tasks.
>
> Table 1 summarizes the results:
> - The merged model shows a **clear degradation in ICL capability** (REGBench) compared to the pretrained model, **together with** a drop in downstream performance.
> - After improving ICL capability via prompt tuning, the model achieves **consistent gains on REGBench** and **noticeable improvements** across downstream tasks.
>
> Overall, these results suggest that **manipulating ICL capability** (while controlling for task knowledge and model parameters) leads to **corresponding changes** in downstream performance, providing **stronger empirical support** for a correlation connection between ICL capability and task conflict.
>
> **Table 1. Intervention study on ICL capability and downstream performance.**
>
> | Model | REGBench (↑) | ANLI (↑) | MCTACO (↑) | COQA (↑) | MATHQA (↑) |
> |---|---|---|---|---|---|
> | Pretrained | 0.864 | 0.414 | 0.627 | 0.420 | 0.345 |
> | Individual | — | 0.634 | 0.953 | 0.630 | 0.417 |
> | Merged (TA) | 0.726 | 0.611 | 0.734 | 0.313 | 0.281 |
> | Merged + Prompt Tuning | 0.785 | 0.655 | 0.702 | 0.371 | 0.294
>
> ---
>
> ### **W2 & Q2. Computational Cost**
> We thank the reviewer for raising concerns about computational cost. We will add a dedicated paragraph reporting the compute trade-off and running time in the final version. Below we give a transparent back-of-the-envelope estimate of OPIC in **GPU-hours**, using a **3B** parameter model on a single **A100** GPU.
>
> We consider a typical ICL benchmark with **D = 40** samples and sequence length **S = 24**, resulting in approximately **960 tokens** per evaluation. Assuming a conservative inference throughput of **150 tokens/s**, each evaluation takes about **6.4 seconds**, i.e., **0.0018 GPU-hours**. Setting the evolutionary search of **G = 20** generations and population size **K = 15** (i.e., **E = 300 evaluations**). Since the sample size on the test set is larger and the text length is longer, we assume that the other algorithms take 0.01 hours for one round of evaluation. The total cost is summarized below:
>
> **Table 2. Estimated computational cost.**
>
> | Method | Eval (E) | Cost/Eval (h) | Total (GPU h) | Remarks |
> |---|---:|---:|---:|---|
> | TA / DARE / TIES | 1 | ~0 | ~0 | One-shot merging |
> | TA / DARE / TIES | 20 | 0.01 | ~0.2 | Grid-search tuning |
> | **OPIC (Ours)** | **300** | **0.0018** | **~0.54** | Evolutionary optimization |
>
> **Conclusion:** Overall, the overall GPU time of OPIC is about **2–3×** tuning-based baselines. However, OPIC **removes the need for task-specific labeled validation data** and provides a **more automated and general** merging strategy.
>
> ---
>
> ### **W3 & Q3. Limited Task Diversity**
>
> Thank you for raising this important concern. We agree that scaling merging to a larger number of tasks is a meaningful direction (we also briefly discuss this in the conclusion).
>
> Our current experiments follow the common setup in closely related work and focus on merging **four representative tasks** to validate the effectiveness of the proposed method. However, larger-scale merging introduces additional scientific questions, such as how conflict differs between **same-type tasks** versus **different-type tasks**, and how these conflict patterns affect merged-model behavior. We view this as an important future direction and plan to conduct larger-scale merging experiments in follow-up work.
>
> ---
>
> ### **W4. Overly Strong Causal Wording**
>
> Thank you for pointing out that some statements in the paper may overstate the empirical evidence. In the final version, we will revise the wording to more accurately reflect our findings—for example, changing **“main cause”** to **“key factors”**—to avoid over-claiming causality.

---

> > ### Author Rebuttal · Reviewer_5PXc · 2026-04-02
> >
> > The authors' rebuttal has addressed all of my concerns. On the causality issue, the additional controlled experiment alleviates my doubts about the credibility of the paper's core hypothesis. It is appropriate that the authors revise the original strong causal claim to a correlation-based interpretation. Overall, I acknowledge the paper's key findings and methodological innovation, and believe it meaningfully improves the practicality and usability of model merging approaches. Therefore, I am inclined to raise my score to 5.

---

> > > ### Author Response · Authors · 2026-04-05
> > >
> > > Thank you very much for your positive feedback and for recognizing the value of the additional controlled experiment. In the final version, we will incorporate this change explicitly and consistently throughout the paper, and we will include the new controlled experiment to better support the paper’s central empirical finding. We are grateful that you found the revised presentation and evidence sufficient to address your concerns.

---

### Official Review · Reviewer_LFHA · 2026-03-11

**Soundness:** 2
**Presentation:** 2
**Significance:** 2
**Originality:** 3
**Overall Recommendation:** 3
**Confidence:** 4

**Summary:**

The authors observe a correlation between the accuracy of the merged model and its in-context learning performance. They therefore propose a new model merging method based on an evolutionary algorithm to find the optimal merging parameters using the in-context learning capabilities as fitness function. They benchmark their method against popular merging methods and find that it performs slightly better. They also provide an ablation of their method, showing its robustness and justifying their design choices.

**Compliance With Llm Reviewing Policy:**

Affirmed.

**Final Justification:**

While I think reworking the text and wording to better reflect the paper's contributions is good and honest, and I appreciate the added experiments, I still do not think the contributions of this work warrant acceptance.

**Key Questions For Authors:**

- There are multiple typos throughout the paper "has -> have achieved" (line 44); \citep -> ~\citep throughout to leave a space between the citation and the preceding word; "Previous" "previous" (line 58); "both matrices B and C" -> "A and B" (line 414-415); etc.
- In the ablation studies, running OPIC without A & B (i.e. only C the global scaling parameter) yields significantly worse results (76.4%) than Task Arithmetic (79.56%) despite both only tuning the scaling hparam. The authors don't discuss this.
- How many steps of the evolution algorithm are necessary to arrive at the final merged models? How does it compare to the number of evaluation steps required by TA or TIES for example?

**Limitations:**

The potential negative societal impact of the work is discussed but the limitations aren't properly discussed in my opinion. As previously mentioned, the authors have misleading claims about their use of data and the computational requirements of their methods which might be significant isn't discussed.

**Strengths And Weaknesses:**

### Strengths
- The identification of in-context learning capabilities as a surrogate for model performance is an interesting idea, and using it to direct merging design can be powerful (Reason I gave 3 to Originality).
- The empirical results are quite extensive, spanning multiple models and methods.
- The figures and tables are clean.

### Weaknesses
- The authors say that the degradation of In-Context Learning capabilities was identified as a primary "driver" of task conflict and model merging performance degradation. In the conclusion they even say that loss of ICL capability is identified as the primary "cause" of task conflict in model merging. The authors employ language that is suggestive of causality, while they only actually observe correlation between ICL capability and model performance (Fig. 2). This is misleading and incorrect, more experimentation / justification is needed to establish a causal link. I believe the idea of using the surrogate metric of ICL capability to predict merging performance is interesting but the paper needs to be re-framed in this way instead of suggesting causality.
- The authors also claim that their method is "validation-free" or "data-free" which is also misleading. Their method does require data, and not only does it require data, the data isn't simply used for hyper-parameter tuning (e.g. as in DARE, TA, TIES) but to optimize the actual training recipe which is an even more involved process. The fact that the data is synthetic or that it isn't directly "test" data on the actual model capabilities doesn't remove from this fact.
- The writing is loose throughout the paper and isn't adequate for a scientific publication. The writing needs to be more precise to avoid misleading the readers (see the weaknesses above) and stating things that are simply not true. Throughout the paper there are issues about the claims, the justifications and even the presentation of related works. For example the authors write "task conflicts from the perspective of parameter alignment, which has been proved not the primary causes (Wang et al., 2024)" but the paper referenced doesn't contain a single proof and how the authors conclude this from the (Wang et al., 2024) paper is unclear to me. This is in part why I gave 2 to Presentation.
- The split of the merging literature into test-time adaptation, MoE-like and Data-free doesn't really make sense since those aren't mutually exclusive. For example some MoE-like methods are Data-free (see MoErging literature). Also AdaMerging isn't exactly "test-time adaptation" since data is used to determine the training recipe but then, at test-time, the merged model is fixed.
- The empirical results are only slightly better than the considered benchmarks, despite OPIC using data to find the actual merging recipe. In this sense, OPIC is closer conceptually to AdaMerge to which the authors do not compare. A fair comparison with AdaMerging and other methods which use data during the merging regime design is necessary for OPIC since it is more similar to these methods than to Task Arithmetic, DARE or TIES who only use data for hparam selection. Because of this, the "Validation Req." column in Table 1 is also misleading, OPIC requires data as well, just different data. This is in part why I gave 2 in Soundness and Significance.
- The computational requirements of OPIC aren't discussed at all but I suspect them to be significant. The authors compare against TA, DARE, TIES which require hparam tuning (but the original papers also suggest some good hparam values that can be used "out of the box") but their method requires multiple optimization steps and evaluation on the ICL benchmark. It's not clear to me that this is a direct win in terms of computational requirements.

My overall opinion of this work is that utilizing validation on an ICL task instead of one of the actual "expert" tasks on which the models were trained is an interesting idea. But the causal link the authors claim is unjustified, and the framing of their method as "data" or "validation"-free is incorrect, it's simply *other type of data* that is used for the validation / tuning. Furthermore the empirical results are quite weak, even when compared to TA, DARE, TIES which don't require extensive optimization through an evolutionary algorithm and the authors don't compare against methods like AdaMerge which also use data to find the merging recipe (a fair comparison in my opinion).

---

> ### Author Rebuttal · Authors · 2026-03-30
>
> We sincerely thank the reviewer for the detailed and thoughtful feedback, and are encouraged by your interest in the potentially impactful core intuition. Below we respond to your concerns.
>
> ---
>
> ## **W1. Causality vs. Correlation**
>
> We thank the reviewer for the recognition and insightful comment. We agree that our current evidence does not support a strong causal claim, and that terms such as *“primary driver/cause”* may be misleading. We will revise the paper to remove such wording and present ICL as a **correlated and useful surrogate signal**.
>
> To strengthen the analysis, we conduct a **controlled experiment**: after merging tasks with Task Arithmetic (Qwen1.5-7B), we improve ICL via soft prompt tuning (without changing model parameters or task knowledge), and evaluate on REGBench and downstream tasks.
>
> ### Table 1: Intervention study
>
> | Model | REGBench ↑ | ANLI ↑ | MCTACO ↑ | COQA ↑ | MATHQA ↑ |
> | --- | --- | --- | --- | --- | --- |
> | Pretrained |0.864|0.414|0.627|0.420|0.345|
> | Merged (TA)|0.726| 0.611|0.734|0.313|0.281|
> | + Prompt Tuning |0.785|0.655|0.702|0.371|0.294|
>
> **Observations and Conclusion:**
> Merging reduces ICL and downstream performance; restoring ICL improves both, providing stronger evidence for a correlation link.
>
> ---
>
> ## **W2. “Validation-free / Data-free” Terminology**
>
> We appreciate the reviewer raising this important inappropriate expression. We will revise the wording to **“no task-specific validation data”** or **“task-label-free”** to more accurately reflect this setting.
>
> ---
>
> ## **W3. Writing Clarity and Related Work Positioning**
>
> We thank the reviewer for noting issues regarding writing precision and related work interpretation. The citation shows that **"weight interference"** is not the primary cause of performance degradation in task arithmetic-based merging. However, in our manuscript, we incorrectly used the term **“parameter alignment”** to refer to this concept leanding to mistake. We will thoroughly revise the manuscript for scientific clarity and audit all citations to ensure faithful representation of prior work.
>
> ---
>
> ## **W4. Categorization of Related Work**
>
> We thank the reviewer for the comment. We agree that our previous taxonomy (test-time adaptation / MoE-like / data-free) was **not strictly orthogonal**, leading to confusion. We will incorporate the reviewer’s suggestions and carefully revise the Related Work section in the final version.
>
> ---
>
> ## **W5. Comparison with AdaMerging**
>
> We agree that AdaMerging is a conceptually relevant comparison because it also learns merging coefficients from data-driven signals. Our current experimental section focused on methods that do not require target-task test samples, but we acknowledge that this scope was not made sufficiently explicit. We will revise the paper to clarify this scope and include following experiment(Settings are the same as the main experiment).
>
> ### Table 2: Additional Comparisons
>
> | Model | ANLI ↑ | MCTACO ↑ | COQA ↑ | MATHQA ↑ |
> |---|---|---|---|---|
> |OPIC|0.618|0.738| 0.410| 0.346|
> |AdaMerging|0.582| 0.572| 0.443 | 0.342 |
>
> **Conclusion:**
> Our approach remains reasonably competitive.
>
> ---
>
> ## **W6 & Q3. Computational Cost and Evolution Steps**
>
> We thank the reviewer for raising this concern. We will include a dedicated discussion of computational cost in the final version. Below is a concise estimate.
>
> Under a typical ICL setting ($Samples D=40$, $ Sequence length S=24$, ≈960 tokens), each evaluation takes **≈0.0018 GPU hours** (150 tokens/s). With $G=20$ generations and population size $K=15$, OPIC requires ~300 evaluations. For baselines, we estimate ~0.01 GPU hours per evaluation.
>
> ### Table 3: Estimated computational cost
>
> |Setting|GPU Cost| Wall-clock(measured in our runs)|
> |---| --- | ---|
> |**OPIC (Ours)** | $\approx 0.54$ GPU hours | $\approx 4.5$ hrs |
> |**Baselines (with tuning)** |$\approx 0.2$ GPU hours  | $\approx 2$ hrs   |
>
> ---
>
> ## **Q1. Typos and Presentation Issues**
>
> We thank the reviewer for identifying typos and formatting issues. We will **thoroughly proofread** the manuscript and correct them.
>
> ---
>
> ## **Q2. Ablation Result  Analysis**
>
> We thank the reviewer for this insightful observation. We agree that the **C-only (single-$\lambda$) setting underperforms Task Arithmetic** and attribute this to three factors:
>
> (1) **REGBench stochasticity** (generation/decoding noise);
> (2) **high sensitivity to $\lambda$** (e.g., 0.30 → 0.35 drops 79.6 → 76.9);
> (3) **ICL is not a complete linearly proxy**, ICL and downstream performance are not strictly linearly aligned.
>
> In the revision, we will provide a more detailed discussion of this phenomenon. Importantly, this further motivates the need for multi-level optimization (A+B+C) in OPIC, which improves robustness and stability compared to single-parameter settings.
>
> ### Table 4: $\lambda$-only sensitivity
>
> | $\lambda$ | Avg. Perf (%) | ICL|
> | --- | --- | --- |
> | 0.25 | 78.5 | 0.872 |
> | 0.30 | 79.6| 0.861 |
> | 0.35  | 76.9 | 0.857 |
> ---

---

> > ### Author Rebuttal · Reviewer_LFHA · 2026-04-01
> >
> > I thank the authors for carefully addressing my main concerns. While I think reworking the text and wording to better reflect the paper's contributions is good and honest, and I appreciate the added experiments, I still do not think the contributions of this work warrant acceptance. I have increased my score to a 3.

---

> > > ### Author Response · Authors · 2026-04-05
> > >
> > > Thank you for your careful reassessment and for acknowledging both the revised wording and the additional experiments. We understand and respect your remaining concern. In the final version, we will continue to present the work in a more restrained and precise way: we will avoid overstating the scope of the claims, clearly separate what is supported by the current experiments from what remains a limitation, and frame OPIC as a practical and empirically supported approach for validation-free model merging rather than a definitive or broadly universal solution. We appreciate this feedback, as it helps us present the work more honestly and clearly.
> > >
> > > In addition, combined with the opinions of the reviewer UZxM, we have also added additional experiments on the **generalization** of the method, hoping to help you understand the contribution of this article more comprehensively.
> > >
> > > ### Experimental Setup:
> > > We construct one merging setting in the **math reasoning** domain using **MathQA, ASDiv, and DROP**, and another in the **logic reasoning** domain using **ANLI, LogiQA, and ARC**. We then evaluate the merged models on both the corresponding **in-domain tasks** and on **MMLU** as an **out-of-domain** test set. This experiment is intended to examine whether the proposed method can still maintain stronger generalization performance when the merged tasks come from relatively specialized domains.
> > >
> > > ### Table 1. Domain generalization results on specialized reasoning domains
> > >
> > > | Setting               | Method            |   MathQA (↑) |    ASDiv  (↑) |     DROP  (↑) | MMLU (OOD)  (↑) |
> > > | --------------------- | ----------------- | -------: | -------: | -------: | ---------: |
> > > | Math reasoning merge  | Task Arithmetic   | 0.4385|  0.5670 |   0.2222 |   0.5476 |
> > > | Math reasoning merge  | TIES               |   0.4358 |  0.5219 |   0.1673 |   0.5903 |
> > > | Math reasoning merge  | DARE             |   0.4458 |  0.5414 |   0.2817 |   0.5880 |
> > > | Math reasoning merge  | CABS             |   0.3986 |  0.3943 |   0.1395 |   0.5957 |
> > > | Math reasoning merge  | DOGE            |   0.4294 |  0.5757 |  0.2960 |    0.5924 |
> > > | Math reasoning merge  | OPIC (ours)    | 0.4494 | 0.5625 | 0.3357 |  0.6169 |
> > >
> > > | Setting     | Method            |     ANLI  (↑) |   LogiQA  (↑) |   ARC  (↑) | MMLU (OOD)  (↑) |
> > > | --------------------- | ----------------- | -------: | -------: | -------: | ---------: |
> > > | Logic reasoning merge | Task Arithmetic   |    0.658 |   0.2457 |   0.7091 |   0.4325 |
> > > | Logic reasoning merge | TIES              |   0.530 |   0.2918 |   0.7739 |   0.5833 |
> > > | Logic reasoning merge | DARE              |    0.573 |   0.2795 |   0.7731 |    0.5700 |
> > > | Logic reasoning merge | CABS                |   0.438 |   0.2887 |    0.7558 |   0.5696 |
> > > | Math reasoning merge  | DOGE              |    0.373 |   0.2918 |   0.7819 |   0.5039 |
> > > | Logic reasoning merge | OPIC (ours)       | 0.663 | 0.2903 | 0.7637 |  0.5933 |
> > >
> > >
> > > ### Observation and Conclusion:
> > >
> > > These results show that OPIC consistently achieves stronger performance not only on the **specialized in-domain** merged tasks, but also on the **out-of-domain** MMLU evaluation, which may be due to the optimization goal of our approach with large model characteristics. These results show that the proposed ICL-oriented optimization objective transfers beyond the specific domains used for merging. We will include this experiment and the corresponding discussion in the final version to make the generalization boundary of the method clearer and better supported.

---

### Official Review · Reviewer_UZxM · 2026-03-12

**Soundness:** 2
**Presentation:** 3
**Significance:** 2
**Originality:** 3
**Overall Recommendation:** 4
**Confidence:** 3

**Summary:**

The paper addresses the challenge of task conflict in task-vector-based model merging, where combining multiple fine-tuned models often leads to performance degradation. The authors argue that current mitigation strategies are overly dependent on external validation sets for hyperparameter tuning. They propose OPIC, a framework that shifts the focus from parameter-level alignment to the preservation of In-Context Learning (ICL) capabilities. A broad area examined by this article is the optimization of model merging without relying on ground-truth task data. The authors use REGBench, a synthetically generated dataset, to serve as a proxy for evaluating ICL retention. They formulate the merging process as an optimization problem using a hierarchical refinement operator and solve it via evolutionary algorithms. Empirically, OPIC achieves 80.73% average performance retention, outperforming SOTA methods by up to 11.1% on benchmarks like ANLI and MathQA.

**Compliance With Llm Reviewing Policy:**

Affirmed.

**Final Justification:**

The rebuttal partially addressed my concerns so I decided to raise my score.

**Key Questions For Authors:**

1. **Task Alignment:** Does REGBench need to be similar to the downstream task to be effective? Is there any guarantee the method works if the specific task is irrelevant to or missing from REGBench?
2. **Hyperparameter Trade-offs:** Does OPIC simply replace the tuning of $\lambda$ with a different set of hyperparameters?
3. **Causality vs. Correlation:** Have you conducted experiments where ICL is preserved but task-specific performance still drops to determine if ICL is a sufficient condition for merging?
4. **Computational Benchmarking:** How does the wall-clock time of the evolutionary optimization in OPIC compare to a standard grid search on a small validation set?
5. **Domain Generalization:** For highly specialized tasks (e.g., medical or legal code), does the general-purpose REGBench suffice, or must the synthetic data generation be domain-aware?
6. **Sensitivity Analysis:** How sensitive is the final model performance to the specific hyperparameters of the evolutionary algorithm itself?

**Minor**:

- Fix typos and formatting

**Limitations:**

The authors should discuss about the computational costs of the optimization process and the potential failure modes if the synthetic data distribution shifts significantly from the task distribution.

**Strengths And Weaknesses:**

**Strengths**

- **Empirical Discovery:** Identifies an interesting correlation between In-Context Learning (ICL) degradation and task conflict during model merging.
- **Practical Relevance:** Addresses the highly relevant goal of reducing dependency on validation data, which is a major bottleneck for practical deployment.
- **Originality:** Introduces a creative application by using ICL as a proxy objective for model merging and combining evolutionary algorithms with synthetic data generation.
- **Presentation Quality:** The paper is generally well-structured and includes a helpful workflow diagram (Figures 1 and 3).

----

**Weaknesses**

- **Lack of Causal Proof:** The causal link between ICL and task conflict is not rigorously established, only observed as a correlation.
- **Efficiency Concerns:** The reliance on evolutionary optimization raises questions about computational efficiency compared to standard merging techniques like TIES or DARE.
- **Overstated Claims:** The "validation-free" claim is slightly overstated, as the method replaces task-specific labels with a dependency on the quality and representativeness of the REGBench synthetic data.
- **Domain Sensitivity:** There is a risk of sub-optimal merging if the synthetic REGBench data fails to capture some specific tasks.

---

> ### Author Rebuttal · Authors · 2026-03-30
>
> We sincerely thank the reviewers for recognizing the contributions of our work—**particularly the interesting finding on the relationship between ICL capability and task conflict, as well as its practical advantages for deployment**. Below, we address your comments regarding **causality, computational efficiency**, and other aspects.
>
> ---
>
> ## **W1 & Q3. Causality vs. Correlation**
>
> We thank the reviewer for pointing out the **imprecise causal wording**. Our current evidence supports a positive correlation between ICL retention and merged-model performance, rather than a definitive causal claim. Based on other reviewers' suggestions and careful reflection, we believe it would be better to revise the causal statement to correlation.
>
> Additionally, you express concern about whether ICL is a sufficient condition for merging. In order to demonstrate this point and provide a more adequate experimental argument for the core idea of correlation,  we conduct following **controlled experiments**:
>
> * Merge four tasks using **Task Arithmetic** on **Qwen1.5-7B**.
> * Apply **Soft Prompt Tuning** on the merged model with **DROID** (unrelated to merging tasks) to improve ICL capability **without changing model parameters or task knowledge**.
> * Evaluate on **REGBench** and downstream tasks.
>
> ### Table 1: Intervention study on ICL capability and downstream performance
>
> | Model| REGBench ↑ | ANLI ↑ | MCTACO ↑ | COQA ↑ | MATHQA ↑ |
> | ---------- | ---------- | ----- | ------ | ----- | ------ |
> | Pretrained  | 0.864 | 0.414 | 0.627  | 0.420 | 0.345  |
> | Merged (TA) | 0.726  | 0.611 | 0.734  | 0.313 | 0.281  |
> | Merged + Prompt Tuning | 0.785| 0.655 | 0.702  | 0.371 | 0.294  |
>
> **Observations and Conclusion:**
> Model merging indeed leads to a degradation of ICL capability. Moreover, improving this capability—without modifying the model parameters—can recover downstream performance and alleviate task conflicts, providing stronger evidence for a **correlation between ICL and merging performance**.
>
> ---
>
> ## **W2 & Q3. Computational Cost**
>
> We thank the reviewer for raising concerns about computational cost. We will add a dedicated paragraph reporting the compute **trade-off and running time** in the final version. Below we give a transparent back-of-the-envelope estimate. We consider a typical ICL setting with $D=40$ samples and sequence length $S=24$ (≈960 tokens per evaluation). Assuming 150 tokens/s, each evaluation takes **≈0.0018 GPU hours**. With an evolutionary search of $G = 20$ generations and population size $K = 15$. Since the sample size on the test set is larger and the text length is longer, we assume that the other algorithms take 0.01 hours for one round of evaluation. The total cost is summarized below:
>
> ### Table 2: Estimated computational cost
>
> | Setting| GPU Cost / Computation|Wall-clock (measured in our runs)|
> | ----- | ----- | ----- |
> | **OPIC (Ours)**| $300 \times 0.0018 \approx 0.54$ GPU hours | $\approx 4.5$ hours|
> | **Baselines (with tuning)** | $20 \times 0.01 \approx 0.2$ GPU hours| $\approx 2$ hours|
>
> ---
>
> ## **W3. “Validation-free” Claim**
>
> Thank you for highlighting this ambiguity. We agree that our current wording may be misleading. We will revise the terminology to:
>  **no task-specific validation data** to more accurately reflect our setting.
>
> ---
>
> ## **W4, Q1 & Q5. Domain Generalization**
>
> Thank you for raising this important concern. The reviewer suggests that **REGBench may need to be closely aligned with downstream domains** to be effective. Therefore, we provide two arguments hoping to eliminate your concerns.
>
> * **Generation of REGBench**
>   REGBench is generated automatically and consists of **structured synthetic sequences**, which **do not encode domain-specific knowledge**.
>
> * **Mechanism of REGBench**
>   REGBench serves as a **proxy objective for evaluating ICL capability**. It leverages the **correlation between ICL capability and task conflict** to estimate downstream performance rather than directly assessing, that ensures the **generalization ability.
>
> ---
>
> ## **Q6. Sensitivity to Evolutionary Algorithm Hyperparameters**
>
> We sincerely thank the reviewer for this insightful observation regarding the sensitivity of evolutionary algorithms (EA) to hyperparameters. From a theoretical perspective, these hyperparameters primarily influence **convergence speed**, and as long as **ICL capability is properly evaluated and optimized**, performance differences should remain limited.
>
> To validate this, we conduct additional experiments analyzing the sensitivity of OPIC to EA hyperparameters. Results are shown below:
>
> ### Table 3: Sensitivity analysis of evolutionary hyperparameters
>
> | Population (K) | Mutation (m) | Performance (%) |
> | -------------- | ------------ | --------------- |
> | 15             | 0.05         | 80.78           |
> | 15             | 0.10         | 80.73           |
> | 20             | 0.05         | 80.55           |
> | 20             | 0.10         | 80.67           |

---

> > ### Author Rebuttal · Reviewer_UZxM · 2026-04-03
> >
> > Thank you to the authors for carefully answering my questions. I appreciate the clarifications provided in the rebuttal, which address some of my initial concerns.
> >
> > After reading the other reviews, I also noticed several common concerns raised across reviewers, including causality, computational cost, the validation-free claim, and domain generalization. I appreciate the additional experiments on causality and computational cost, as well as the authors’ explanations regarding the validation-free setting and domain generalization. However I feel these points somewhat weaken the significance of this paper. In addition, I am still interested in seeing domain generalization experiments on more specialized or challenging domains, which would help better demonstrate the robustness and broader applicability of the proposed method.
> >
> > Overall, the rebuttal has improved my understanding of the work and addressed part of my concerns. As a result, I will increase my score to 4.

---

> > > ### Author Response · Authors · 2026-04-05
> > >
> > > Thank you for your thoughtful follow-up and for increasing your score. We appreciate that our rebuttal clarified part of your concerns, and we also agree that your emphasis on the need for stronger generalized evidence is important. To further address this concern, we added a new domain generalization experiment covering two specialized reasoning domains.
> > >
> > > ### Experimental Setup:
> > > We construct one merging setting in the **math reasoning** domain using **MathQA, ASDiv, and DROP**, and another in the **logic reasoning** domain using **ANLI, LogiQA, and ARC**. We then evaluate the merged models on both the corresponding **in-domain tasks** and on **MMLU** as an **out-of-domain** test set. This experiment is intended to examine whether the proposed method can still maintain stronger generalization performance when the merged tasks come from relatively specialized domains.
> > >
> > > ### Table 1. Domain generalization results on specialized reasoning domains
> > >
> > > | Setting               | Method            |   MathQA (↑) |    ASDiv  (↑) |     DROP  (↑) | MMLU (OOD)  (↑) |
> > > | --------------------- | ----------------- | -------: | -------: | -------: | ---------: |
> > > | Math reasoning merge  | Task Arithmetic   | 0.4385|  0.5670 |   0.2222 |   0.5476 |
> > > | Math reasoning merge  | TIES               |   0.4358 |  0.5219 |   0.1673 |   0.5903 |
> > > | Math reasoning merge  | DARE             |   0.4458 |  0.5414 |   0.2817 |   0.5880 |
> > > | Math reasoning merge  | CABS             |   0.3986 |  0.3943 |   0.1395 |   0.5957 |
> > > | Math reasoning merge  | DOGE            |   0.4294 |  0.5757 |  0.2960 |    0.5924 |
> > > | Math reasoning merge  | OPIC (ours)    | 0.4494 | 0.5625 | 0.3357 |  0.6169 |
> > >
> > > | Setting               | Method            |     ANLI  (↑) |   LogiQA  (↑) |   ARC  (↑) | MMLU (OOD)  (↑) |
> > > | --------------------- | ----------------- | -------: | -------: | -------: | ---------: |
> > > | Logic reasoning merge | Task Arithmetic   |    0.658 |   0.2457 |   0.7091 |   0.4325 |
> > > | Logic reasoning merge | TIES              |   0.530 |   0.2918 |   0.7739 |   0.5833 |
> > > | Logic reasoning merge | DARE              |    0.573 |   0.2795 |   0.7731 |    0.5700 |
> > > | Logic reasoning merge | CABS                |   0.438 |   0.2887 |    0.7558 |   0.5696 |
> > > | Math reasoning merge  | DOGE              |    0.373 |   0.2918 |   0.7819 |   0.5039 |
> > > | Logic reasoning merge | OPIC (ours)       | 0.663 | 0.2903 | 0.7637 |  0.5933 |
> > >
> > >
> > > ### Observation and Conclusion:
> > >
> > > These results show that OPIC consistently achieves stronger performance not only on the **specialized in-domain** merged tasks, but also on the **out-of-domain** MMLU evaluation, which may be due to the optimization goal of our approach with large model characteristics. These results show that the proposed ICL-oriented optimization objective transfers beyond the specific domains used for merging. We will include this experiment and the corresponding discussion in the final version to make the generalization boundary of the method clearer and better supported.

---

### Decision · Program_Chairs · 2026-04-30

**Decision:**

Accept (regular)

**Comment:**

This paper proposes OPIC, a framework that formulates model merging as an optimization problem aimed at preserving In-Context Learning (ICL) capabilities, thereby reducing reliance on task-specific validation data. The reviewers broadly agree on the paper's novelty in identifying a correlation between ICL degradation and task conflict, as well as the practical utility of a "validation-free" approach for deployment. However, significant concerns were raised regarding the authors' initial overstatement of causality (claiming ICL loss is a "primary cause" rather than a correlated factor) and the ambiguity surrounding the "data-free" claim given the reliance on synthetic REGBench data. Additionally, Reviewer LFHA questioned the computational efficiency of the evolutionary algorithm compared to standard baselines and noted issues with the related work categorization.

Following the rebuttal, the authors have satisfactorily addressed the majority of these concerns. They have revised the manuscript to reframe their claims from causal to correlational, supported by new intervention experiments demonstrating that improving ICL capability recovers downstream performance. The authors also provided detailed computational cost estimates showing OPIC incurs approximately 2–3x the cost of grid-search baselines, which is deemed acceptable given the automation benefits, and added comparisons with AdaMerging to clarify the related work landscape. Given the technical solidity, the novelty of the ICL-based optimization objective, and the successful resolution of major concerns regarding causality and efficiency, I recommend acceptance, contingent on the authors incorporating the suggested clarifications and experiments in the final version.